# Extracting Deformation-Aware Local Features by Learning to Deform

**Guilherme Potje** *
Universidade Federal de Minas Gerais

**Renato Martins**
Université Bourgogne Franche-Comté

**Felipe Cadar**
Universidade Federal de Minas Gerais

**Erickson R. Nascimento**
Universidade Federal de Minas Gerais

## Abstract

Despite the advances in extracting local features achieved by handcrafted and learning-based descriptors, they are still limited by the lack of invariance to non-rigid transformations. In this paper, we present a new approach to compute features from still images that are robust to non-rigid deformations to circumvent the problem of matching deformable surfaces and objects. Our deformation-aware local descriptor, named DEAL, leverages a polar sampling and a spatial transformer warping to provide invariance to rotation, scale, and image deformations. We train the model architecture end-to-end by applying isometric non-rigid deformations to objects in a simulated environment as guidance to provide highly discriminative local features. The experiments show that our method outperforms state-of-the-art handcrafted, learning-based image, and RGB-D descriptors in different datasets with both real and realistic synthetic deformable objects in still images. The source code and trained model of the descriptor are publicly available at `https://www.verlab.dcc.ufmg.br/descriptors/neurips2021`.

## 1 Introduction

High-quality matching of images taken from cameras in different poses and conditions remains one of the key challenges in several tasks. Tracking points as a camera or an object of interest is moving is crucial for finding and reconstructing objects and scenes [29, 30, 31], camera localization and mapping [18, 36], registration [43, 37], and image retrieval [26, 39], to name a few tasks. Extracting invariant and compact representations to describe the vicinity of a pixel has been one of the key forces in enabling high-quality matching on these tasks. Despite the importance of invariance to varying illumination, viewpoint, and the distance to the object of interest, real-world scenes impose additional challenges. After all, we live in a non-rigid world populated by objects that may have different shapes over time due to deformations. Over the past few decades, many descriptors have been proposed [19, 3, 6, 35, 42, 47, 44, 40, 23, 13]. These approaches can be roughly grouped based on the type of input information, such as intensity and/or depth images, and can be further divided into handcrafted and learning-based methods. While local learning-based approaches, such as Log-Polar [13], hold state of the art regarding affine transformations and local illumination changes, recent handcrafted techniques achieve better performance in the presence of non-rigid transformations by using depth data [25] or by modeling the deformations from image intensity information, which inevitably leads to an increase of the computational cost to compute the description [38]. Since depth data is not available in many contexts and applications, and low processing time is highly

---

*Department of Computer Science, Universidade Federal de Minas Gerais, Brazil.
VIBOT EMR CNRS 6000, ImViA, Université Bourgogne Franche-Comté, France.
Corresponding author's e-mail: `guipotje@dcc.ufmg.br`

35th Conference on Neural Information Processing Systems (NeurIPS 2021).

recommended for low-level processing such as the estimation of local features, devising a descriptor that correctly and reliably establishes invariant features from corresponding points is of central importance to high-quality matching of images in real-world scenarios.

In this paper, we present a new end-to-end method to extract local features from still images that are robust to non-rigid deformations. Conversely to recent approaches that exploit a single network branch to extract features from local patches or depth data, our approach is composed of an additional guidance branch component (spatial transformer non-rigid warper) that jointly learns context about the global deformation affecting the image.

Figure 1 depicts the rationale behind our method. For learning the contextual deformation information, we provide several views of deformed objects by applying isometric non-rigid transformations to the objects in a simulated environment as guidance to extract highly discriminative local deformation-aware features. Unlike non-rigid aware deformations descriptors like GeoBit [25], our approach does not require depth data to produce descriptors robust to nonrigid deformations and it is notably faster than DaLI [38] while holding the best performance.

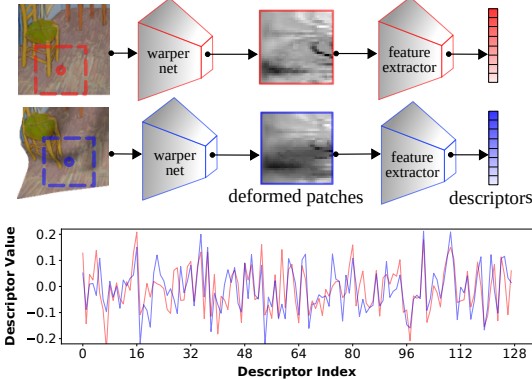

Although many descriptors still disregard nonrigid transformations, in recent years, that assumption has been dispelled, partly due to the increasing popularity in the use of RGB-D sensors to capture properties that characterize 3D objects such as geodesic distances. On the flip side, RGB-D sensors are still less prevalent than RGB cameras. In our nonrigid-world with flexible humans, animals, tissues, and leaves where high-resolution RGB cameras are ubiquitous, the ability to create discriminative and deformation-invariant descriptors using a single picture plays a central role in achieving high-

Figure 1: **Learning to deform and extract features**. Our network learns meaningful deformations during training that lead to improved matching performance, as shown in the experiments. Notice that the descriptor's signatures (Descriptor Value plot) of corresponding keypoints, with and without a local deformation (red and blue curves), are highly correlated when extracted with the proposed model.

quality matching in real-world scenarios. In this work, we demonstrate that our end-to-end architecture outperforms state-of-the-art handcrafted and learning-based image and RGB-D descriptors in matching scores on different benchmark datasets of real deformable objects, as well as with realistic application scenarios on content-based object retrieval and tracking.

## 2 Related work

**Handcrafted and learning-based feature extraction.** Traditionally, image descriptors are aimed to providing rich and invariant representations of objects observed from different viewing conditions. Two notable handcrafted approaches to describe keypoints on still images are SIFT [19] and ORB [35]. These methods use local gradients and dominant orientation estimation to extract visual features with scale and rotation invariance. Different extensions of these descriptors have been proposed to images in some particular manifolds such as BRISKS [14] and SphORB [50] for spherical images. Recently the advent of convolutional neural networks boosted the development of several end-to-end learned descriptors [47, 44, 40, 23, 13, 20, 32]. These descriptors often outperform handcrafted counterpart on classical image matching benchmarks such as UBC Phototour [5] and HPatches [2]. Yet, the vast majority of existing local descriptors are approximately invariant to affine image transformations, often disregarding images of deformable surfaces, which are the main feature of our descriptor.

**Deformation-aware reconstruction and matching.** Although most works on local descriptors are designed focusing on affine image transformations, some works considered non-rigid surfaces and deformable objects. In DeepDeform [4], for instance, the authors propose to estimate non-rigid point-wise RGB-D correspondences by a data-driven approach that is then used by a non-rigid 3D reconstruction framework. Although their method works well for fast motions and non-rigid surface

tracking, it has a heavy computational burden for establishing correspondences for typical amounts of keypoints from commonly used detectors. Also, the method does not generate a descriptor that can be used for other related vision tasks. Simo-Serra *et al.* proposed the DaLI descriptor [38], where image patches are converted to a 3D surface in order to encode features robust to non-rigid deformations and illumination changes. Despite achieving high matching performance, DaLI suffers from high computational and storage requirements. Recently, DeformNet [31] was designed to infer non-rigid shapes from still images. The network architecture is composed of a 2D detection branch and a depth branch. The detection and depth branches are subsequently merged by the shape branch, which uses a pinhole camera model to re-project the shape to 3D. In the same context, GeoBit [25] takes advantage of geodesics from objects surface to compute isometric-invariant features. The method uses RGB-D data combined with the Heatflow [7] method to encode visual features in a binary string that are invariant to deformations. However, GeoBit requires accurate depth information and only works with RGB-D images. Our method, in its turn, only requires monocular images, is faster to compute, and provides a better matching performance, as demonstrated in our experiments.

**Spatial attention mechanisms and matching.** Spatial transformer networks (STNs), introduced by the seminal work of Jaderberg *et al.* [16], allowed differentiable warping operations to be used directly with existing network architectures. The core idea is the learning of an attention mechanism (named Localization network) that is parameterized to represent the image transformation of interest, *e.g.*, affine, homography or a thin-plate-spline warp. They have been applied to several tasks including image classification [16], semantic alignment [34], geometric matching [33] and local descriptors [13]. LF-Net [27], a detect-and-describe method, first locates interest points using a fully convolutional network, which are then cropped in patches with a STN layer considering an affine transformation encoded by keypoint attributes. The cropped patches are subsequently used to compute local descriptors by the description network. Spatial attention mechanisms have also been extended into more generic settings via deformable kernels [9], to learn kernel offsets in network layers operators to define deformable convolution networks. More recently, the end-to-end joint detector and descriptor ASLFeat [21] used deformable convolutions to increase the network's expressiveness, which shares some concepts with our method. However, the deformable kernels employed by ASLFeat considers high-level feature maps, and it does not model deformations explicitly. Its strategy results in earlier layers not being aware of low level image deformations, usually present in geometric transformations. Moreover, the method is not robust to rotation changes. Our method leverages two carefully designed spatial transformer networks to sample and guide the model to provide invariance to deformations and, to the best of our knowledge, is the first end-to-end learning method to explicitly tackle non-rigid deformations with a carefully designed spatial attention mechanism.

## 3 Deformation-aware network architecture

Our DEformation-Aware Local descriptor (DEAL) has been designed to jointly learn to undeform and extract discriminative and invariant features from local regions. The proposed architecture is able to handle deformations and perspective distortions of sampled patches in the vicinity of the keypoint. It also attenuates noise in keypoint attributes which improves the local description performance and robustness to image transformations. We designed our network as a hybrid approach, in the sense that it first extracts dense features and the final output is a local descriptor for each keypoint, unlike both trends of describing local patches like HardNet [23] or extracting dense features jointly with keypoints such as SuperPoint [10] and R2D2 [32]. Our approach combines the best of these two trends: i) the robustness to occlusion, strong illumination, and perspective changes of local patch-based descriptors that work directly with low-level image information; and ii) the capability of dense methods in encoding local semantics from the image to disambiguate hard pairs. Additionally, our model decouples the problem of learning distinctive descriptors robust to deformations in two complementary tasks, *i.e.*, first estimating a warp with a spatial attention mechanism and then describing the sampled local patches using the estimated non-rigid warp. Figure 2 outlines a schematic representation of the model.

### 3.1 Mid-level feature extraction

For the extraction of mid-level image features that encode local deformation information, we employ the ResNet-34 [15] and truncate it at the sixth convolutional block. The motivations to process

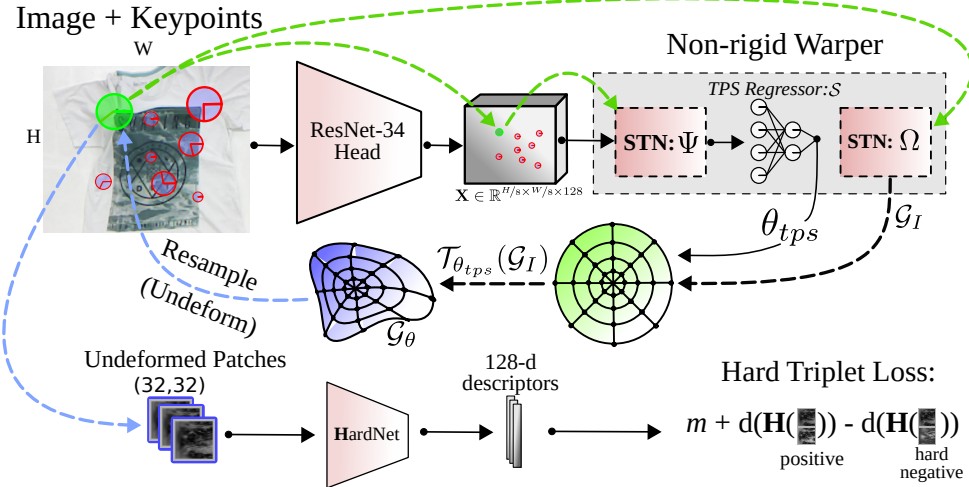

Figure 2: **Proposed formulation for computing descriptors of deforming objects.** The non-rigid warper component undeforms local image patches by applying a learned deformation for each keypoint. Higher level image information such as appearance and shape are encoded globally by the ResNet feature tensor $\mathbf{X}$. Then, the two carefully designed spatial transformers and the TPS regressor network decode the information locally, by estimating the warp parameters $\theta_{tps}$ used to rectify patches from the original image. The green arrows imply keypoint attribute information. The rectified patches are fed to HardNet that extracts discriminant descriptors. The full network model is trained end-to-end by optimizing the hard triplet loss.

features from ResNet instead of directly sampling local patches over the image are twofold: First, the mid-level feature maps provided by ResNet features allow the network to learn to encode local deformations from contextual cues such as local texture statistics, illumination and viewpoint changes; Secondly, the receptive field of our local feature descriptor is not tied to a fixed support region size and sampling pattern like traditional patch-based local descriptors.

Strong perspective and deformation changes drastically influence the optimal support region sizes for the same keypoint on different images; thus, we present an architecture that is aware of these local deformations by design, since the dense feature map allows such information to flow to the non-rigid warper module. The ResNet-34 head outputs a feature map $\mathbf{X} \in \mathbb{R}^{H/8 \times W/8 \times 128}$ by performing convolutions until the spatial resolution achieves $1/8$ of the original input resolution. The feature map $\mathbf{X}$ is used as input for the the non-rigid warper component, which computes the Thin-Plate Splines (TPS) parameters with the spatial transformer to model a non-rigid warp.

## 3.2    Non-rigid warper

The Non-Rigid Warper (NRW) is composed by two spatial transformers $\Psi$ and $\Omega$, in addition to the TPS parameter estimator $\mathcal{S}$. After obtaining the mid-level feature tensor of the entire image $\mathbf{X}$, for each keypoint, $\Psi$ samples a local $5 \times 5 \times 128$ tensor centered at the coordinate of the keypoint, which are flattened and forwarded to $\mathcal{S}$, to estimate the local deformation parameters of the keypoint. Afterward, $\Omega$ initializes the polar transformation $\mathcal{G}_I$ with the keypoint attributes, and finally, patches are re-sampled from the original image and propagated to HardNet descriptor extractor.

**Feature sampler.**    We designed a Spatial Transformer Network (STN) $\Psi$ to sample a local patch of mid-level features $\mathbf{Y}$ from the ResNet feature map $\mathbf{X}$ at the position of each keypoint in an end-to-end differentiable manner. For each keypoint, given a 2D affine transformation function $\mathcal{T}_{\mathbf{K}}$, where $\mathbf{K}$ is an affine matrix obtained from the downscaled keypoint position (to match the spatially downscaled spatial feature map $\mathbf{X}$ by a factor of 8), and the identity mesh grid $\mathcal{M}_I$, we generate transformed mesh grid coordinates as: $\mathcal{M}_{\mathbf{K}} = \mathcal{T}_{\mathbf{K}}(\mathcal{M}_I)$. Once the meshgrid is transformed, for each pixel's spatial coordinate $\mathbf{p}$, $\mathbf{p} \in \mathbb{R}^2$ in $\mathcal{M}_{\mathbf{K}}^{(i,j)}, \forall i \in [1, H], \forall j \in [1, W]$, we bilinearly interpolate values in

$\mathbf{X}$ for each channel $c$ to obtain the warped features $\mathbf{Y}$ per keypoint:

$$\mathbf{Y}_{\mathbf{p}}^c = \sum_{m=0}^{1} \sum_{n=0}^{1} \mathbf{X}_{\lfloor \mathbf{p}+(m,n) \rfloor}^c \mid 1 - m - \mathbf{p}_{(1)}' \mid \times \mid 1 - n - \mathbf{p}_{(2)}' \mid, \qquad (1)$$

where $\mathbf{p}'$ is the decimal part of the pixel coordinate $\mathbf{p}$ [16] and $\mid x \mid$ is the non-negative value of $x$.

The mesh grid $\mathcal{M}_{\mathbf{K}} \in \mathbb{R}^{5 \times 5 \times 2}$ encodes a local patch of 2D spatial coordinates centered at the keypoint. $\mathcal{M}_{\mathbf{K}}$ is used to interpolate the feature map $\mathbf{X}$ to sample the local mid-level features that encode local keypoint deformations. The local feature maps $\mathbf{Y}$ are flattened and forwarded to the localization network that regresses the parameters of the local deformations.

**Localization network.** We model the non-rigid object deformations with Thin-Plate Splines (TPS) [11]. TPS warps representing 2D coordinate mappings $\mathbb{R}^2 \mapsto \mathbb{R}^2$ are often used to model non-rigid deformations. They produce differentiable smooth interpolations, which can be easily coupled in neural networks using attention mechanisms. The TPS formulation elegantly enables one to model the affine component of the transformation through an affine matrix component $\mathbf{A} \in \mathbb{R}^{2 \times 3}$, and a non-affine component encoded by the weight coefficients $\mathbf{w}_k \in \mathbb{R}^2$ separately representing distortions. Given a 2D point $\mathbf{q} \in \mathbb{R}^3$, weight coefficients and control points $\mathbf{c}_k, \in \mathbb{R}^2$ both in homogeneous coordinates, the corresponding mapping $\mathbf{q}'$ can be computed with:

$$\mathbf{q}' = \mathbf{A}\mathbf{q} + \sum_{k=1}^{n} \rho(\|\mathbf{q} - \mathbf{c}_k\|^2)\mathbf{w}_k, \qquad (2)$$

where $\rho(r) = r^2 \log r$ is the TPS radial basis function. To estimate the coefficient weights of the TPS, we employ a regressor network $\mathcal{S}$, implemented as a Multi-layer Perceptron (MLP) that takes as input the interpolated feature map $\mathbf{Y}$ of each keypoint to compute the TPS parameters $\theta_{tps}$, that is composed by the affine parameters $\mathbf{A}$ and the weight coefficients $\mathbf{w}_k$. We use $64$ control points in our implementation, which provides a good trade-off between non-rigid modeling capacity and computational cost according to our experiments. The estimated $\theta_{tps}$, in conjunction with the identity polar grid $\mathcal{G}_I$ containing all control points $\mathbf{c}_k$ (initialized using the Polar Transformer) are used to sample a rectified image patch.

**Spatial transformer network sampler.** According to Ebel *et al.* [13], polar representations of a patch around keypoints increase the robustness of the descriptor extractor to image transformations, while also achieving rotation equivariance. Considering these findings, we adopt this representation in the sampled rectified patches. For that, a fixed regular polar transformation $\mathcal{G}_I$ is computed in the second Spatial Transformer Network $\Omega$, parameterized by the keypoints' attributes. The attributes are obtained from a keypoint detector such as SIFT, and encode the keypoint's location (its coordinates in the image), its canonical orientation (used to achieve image in-plane rotation invariance), and its size (an estimate of the support region around the keypoint). We use those attributes to generate an identity polar transformation map $\mathcal{G}_I \in \mathbb{R}^{32 \times 32 \times 2}$ for each keypoint, encoding the point's initial position, rotation and size. The identity grid $\mathcal{G}_I$ is transformed to the warped grid $\mathcal{G}_\theta$ using the estimated $\theta_{tps}$, which samples the final rectified patches by the second STN $\Omega$. Finally, the patches are forwarded to the feature extraction network that computes a distinctive and compact descriptor for the rectified polar patch. We employed HardNet [23] as the backbone network in our implementation.

### 3.3 Local descriptor extraction and loss

Our architecture can be trained end-to-end since all employed operations are differentiable, including the non-rigid warper component. The HardNet $\mathbf{H}(.)$ takes as input $32 \times 32$ grayscale patches rectified by the non-rigid warper, and outputs a 128-dimensional L2 normalized feature vector for each keypoint. Then, the hardest in-batch triplet loss [23] is computed on a mini-batch of pairs of feature vectors, enforcing distinctiveness of the extracted descriptors.

Let $\mathbf{A} \in \mathbb{R}^{N \times D}$ and $\mathbf{B} \in \mathbb{R}^{N \times D}$ be a matrix of $N$ vertically stacked $D$-dimensional feature vectors of corresponding patches extracted by HardNet. Assuming that the descriptors are L2 normalized, we can calculate the matrix of distances $\mathbf{D}_{N \times N} = 2(1 - \sqrt{\mathbf{A}\mathbf{B}^T})$. The hardest negative example $h_i$ for each row $\mathbf{D}_i'$, $\mathbf{D}' = \mathbf{D} + \alpha\mathbf{I}_{N \times N}$, where $\alpha$ is a constant $\geq 2$ needed to supress the corresponding

pairs from the diagonal of the distance matrix, is computed as $\delta_h^{(i)} = \min(\mathbf{D}_i')$. The hardest in-batch margin ranking loss is then calculated as:

$$\mathcal{L}_H \left( \delta_+^{(\cdot)}, \delta_h^{(\cdot)} \right) = \frac{1}{N} \sum_{i=1}^{N} \max(0, \mu + \delta_+^{(i)} - \delta_h^{(i)}), \tag{3}$$

where $\mu$ is the margin, $\delta_+ = \|\mathbf{H}(p) - \mathbf{H}(p')\|_2$ is the distance between the corresponding patches, and $\delta_h = \|\mathbf{H}(p) - \mathbf{H}(h)\|_2$ is the distance to the hardest negative sample in the batch.

## 4 Experiments and results

We evaluate our descriptor in different publicly available datasets containing deformable objects in diverse viewing conditions such as illumination, viewpoint, and deformation. For that, we have selected the two datasets recently proposed by GeoBit [25] and DeSurT [45]. They contain color images of 11 deforming real-world objects, where keypoints are independently detected with SIFT and ground-truth correspondences are done following the protocol of [17].

**Baselines and metrics.** For the descriptors based on pre-detected keypoints, we use SIFT keypoints that are detected independently for each frame and shared among all methods, and we detect at most $2,048$ keypoints. Since this paper tackles the problem of matching images of deformable objects, we compare our results against two deformation-invariant local descriptors: DaLI [38] and GeoBit [25]. Aside from these descriptors, we also include two the well-known local binary descriptors for 2D images ORB [35] and FREAK [1]; a floating-point gradient based descriptor: DAISY [42]; one local descriptor that combines texture and shape: BRAND [24]; and three learning methods, TFeat [44], Log-Polar [13], which is an improvement of HardNet [23], and SOSNet [41].

We also demonstrate that our method is able to surpass recently proposed detect-and-describe methods. These methods detect their own 2,048 keypoints that are optimized for their descriptors. Since the detect-and-describe techniques do not share the same keypoints, their evaluated scores can be also impacted by the repeatability of their keypoints, and directly comparing them to the local descriptors that use the same set of SIFT keypoints is not straightforward. We have included those methods in the experiments for a more comprehensive evaluation of recent methods. We considered in the comparisons R2D2 [32], ASLFeat [21], D2-Net [12], LF-Net [27], and LIFT [48].

The performance assessment is done with the matching score (MS) metric defined by Mikolajczyk *et al.* [22]. Considering a pair of images $i$ and $j$, the metric is defined as: $MS = \#correct/\min(\#keypoints_i, \#keypoints_j)$, where $\#correct$ is the number of correct keypoint matches at a 3 pixels threshold (used in all experiments in this paper) obtained by matching the descriptors using nearest neighbor search, and $\#keypoints_{\{i,j\}}$ is the number of detected keypoints on image $i$ and $j$. We also compute the Mean Matching Accuracy (MMA @ 3 pixels) as $MMA = \#correct/\#matchable$, where $\#matchable$ is the number of points that can be matched under the threshold.

### 4.1 Implementation details

**Training dataset.** Although most of the reported results are from real-world images, our network was trained with simulated data only. For that we developed a simulation physics engine to generate plausible non-rigid deformations (isometric transformations) of surfaces (please check the supplementary material for additional details of the simulation engine). The dataset is composed of $20,000$ ($960 \times 720$ resolution) image pairs with ground-truth correspondences. In order to generate realistic deformed images *in the simulation engine*, we performed texture mapping of the surfaces with real images extracted from large-scale Structure-from-Motion datasets [46]. We also added illumination variation such as intensity, global position, number of light sources, directional lighting, and color changes to enforce realistic non-linear illumination conditions in the simulation. The correspondences between frames are generated by first independently detecting up to 2,048 SIFT keypoints for each frame and then corresponding them using the simulation data under a threshold of 3 pixels. Each image pair contains, on average, approximately $1K$ ground-truth correspondences, summing up to a total of about $20M$ keypoint correspondences. Independently detecting the keypoints in the frames is a key step to improve generalization, since it considers repeatability properties of keypoint detectors.

**Network and training setup.** We implement [2] our network using PyTorch [28] and optimize it via Adam with initial learning rate of $5e^{-5}$, scaling it by $0.9$ every $3,800$ steps. The network is trained end-to-end, setting the TPS regressor's weights of the last layer to zero, resulting in an identity warp at the beginning of training. We used a batch size of $8$ image pairs containing up to $128$ keypoint correspondences for each pair in our setup. The keypoint correspondences are randomly sampled from a uniform distribution with fixed seed during training, and we train the network for $10$ epochs. The model is trained using a siamese scheme, where descriptors are extracted for the first and second set of corresponding keypoints (positive examples) using two networks with shared weights. The negative examples are calculated using a hard mining strategy in the batch as described by Mishchuk *et al.* [23] and used in the triplet loss. We also apply Average Pooling in the angle-axis of the polar patches at the end of HardNet feature maps to achieve rotation invariance. Our network implementation has $3.7M$ trainable parameters and takes about $5.5$ hours to train on a GeForce GTX 1080 Ti GPU.

**Sensitivity analysis.** To evaluate the sensitivity and the influence of hyperparameters in our network, we performed the following experiments: (i) margin ranking parameter sensitivity in the triplet loss; (ii) use of anchor swap in the triplet loss [44]; (iii) larger support region for the non-rigid warper (NRW) component; and (iv) use of dropout in the fully connected (FC) layers. Table 1 shows the MMA average achieved by testing different hyperparameters on the *Bag* sequence [25], which we used as a validation set and removed it from the benchmark experiments. The training steps and initialization are kept the same for all tested variations, and only one tested hyperparameter is changed while fixing all others for the sake of reproducibility and consistency. The baseline model uses the margin $\mu = 0.5$, no anchor swap, STN output of $3 \times 3$ and Dropout with probability $p = 0.1$ in the FC layers. We can observe that using the margin $0.75$ and STN output of $5 \times 5$ increases the performance individually. We tested both changes simultaneously but it resulted in worse results than the individual changes. Thus, we update the final model (used in all experiments) to use STN output of $5 \times 5$ and keep other parameters from baseline unchanged, since they decrease the performance. We did not include larger STN outputs since we noticed marginal performance gains, while the model increases its computational requirements.

Table 1: Effect of hyperparameters in the proposed network.

| Hyperparameter | MMA $\uparrow$ |
|---|---|
| Baseline | 0.603 |
| Margin 0.25 | 0.592 |
| Margin 0.75 | 0.608 |
| Anchor Swp. | 0.592 |
| STN out. $5 \times 5$ | 0.613 |
| No Dropout FC | 0.601 |

### 4.2 Results on real image sequences of deformable objects

Table 2 shows the matching scores for all descriptors in our experiments. For the sake of a fair comparison, the keypoints are shared among all local descriptors in this experiment, including rotation and scale attributes of SIFT keypoints, with exception of methods that detect their own keypoints jointly with the descriptors. In this case, it is worth recalling that the repeatability of the detector is also impacting the performance of MS and MMA. One can see that our method outperforms all local descriptors that use SIFT keypoints, including GeoBit, that uses additional depth information to extract deformation-invariant features. GeoBit, as expected, achieved the second-best accuracy overall, followed by the learning-based methods. DaLI performs the best among the hand-crafted descriptors, due to its robustness against deformations. BRAND, for its turn, provides the worst scores overall because of its strong assumptions of geometric rigidity of the object in both image and 3D space. LF-Net performs best among the joint detection and description methods. We also re-trained two of the learning-based methods, TFeat** and Log-Polar**, to verify if the proposed non-rigid training dataset can improve the matching of deformable surfaces for existing methods. The re-trained TFeat** on our training data did not improve the MS, and slightly decreased the MMA. On the other hand, the re-trained Log-Polar** yielded an increase from $0.57$ to $0.65$ MMA (8 p.p difference), but it is still 10 p.p. under the performance of our deformation-aware descriptor scores. Additional details about the re-trained methods are available in our supplementary material. Figure 3 shows a matching example from Kinect 1 dataset (*Shirt 1* sequence), depicting a strong deformation. It is noticeable that our method improved the matching, especially in the most deformed regions (red dashed square). We also evaluated our descriptor on HPatches [2], a dataset of planar scenes

---

[2]The training data and source code are available at `www.verlab.dcc.ufmg.br/descriptors/neurips2021`.

Table 2: **Comparison with state-of-the-art descriptors**. The methods marked with $^*$ are computed on RGB-D images. The mark $^{**}$ indicates that we re-trained the networks using our proposed non-rigid dataset, and $^†$ indicates that the method detect its own $2,048$ keypoints. Best in bold and second-best underlined. The mean was calculated with full-precision values before rounding. These results indicate that the proposed descriptor exhibit improved robustness against deformations in all considered datasets.

| Descriptors | Datasets: 833 **pairs total** – MS / MMA @ 3 pixels ↑ | | | | Mean |
| | *Kinect 1* [25] | *Kinect 2* [25] | *DeSurT* [45] | *Simulation* [25] | |
|---|---|---|---|---|---|
| BRAND* [24] | 0.17 / 0.34 | 0.22 / 0.49 | 0.14 / 0.33 | 0.04 / 0.09 | 0.16 / 0.34 |
| R2D2$^†$ [32] | 0.17 / 0.36 | 0.25 / 0.59 | 0.14 / 0.32 | 0.06 / 0.16 | 0.17 / 0.39 |
| ORB [35] | 0.19 / 0.38 | 0.25 / 0.55 | 0.18 / 0.40 | 0.14 / 0.30 | 0.20 / 0.43 |
| SOSNet [41] | 0.17 / 0.34 | 0.25 / 0.55 | 0.17 / 0.38 | 0.26 / 0.57 | 0.22 / 0.47 |
| DAISY [42] | 0.23 / 0.47 | 0.29 / 0.62 | 0.16 / 0.37 | 0.19 / 0.39 | 0.22 / 0.48 |
| FREAK [1] | 0.24 / 0.49 | 0.33 / 0.72 | 0.16 / 0.38 | 0.15 / 0.31 | 0.23 / 0.51 |
| DaLI [38] | 0.25 / 0.51 | 0.35 / 0.76 | 0.21 / 0.48 | 0.10 / 0.22 | 0.25 / 0.54 |
| TFeat** | 0.23 / 0.48 | 0.28 / 0.62 | 0.20 / 0.46 | 0.28 / 0.61 | 0.25 / 0.55 |
| TFeat [44] | 0.25 / 0.50 | 0.28 / 0.61 | 0.21 / 0.48 | 0.29 / 0.63 | 0.26 / 0.56 |
| ASLFeat$^†$ [21] | 0.31 / 0.58 | 0.39 / 0.69 | **0.28** / 0.53 | 0.19 / 0.35 | 0.31 / 0.56 |
| Log-Polar [13] | 0.28 / 0.58 | 0.30 / 0.65 | 0.23 / 0.54 | 0.22 / 0.49 | 0.26 / 0.57 |
| D2-Net$^†$ [12] | 0.20 / 0.50 | 0.23 / 0.82 | 0.14 / 0.47 | 0.11 / 0.30 | 0.17 / 0.57 |
| LF-Net$^†$ [27] | **0.44** / 0.40 | **0.51** / 0.43 | **0.28** / **0.77** | 0.21 / 0.74 | **0.36** / 0.59 |
| LIFT$^†$ [48] | 0.09 / 0.57 | 0.16 / 0.65 | 0.08 / 0.52 | 0.13 / 0.73 | 0.12 / 0.62 |
| Log-Polar** | 0.29 / 0.60 | 0.31 / 0.69 | 0.24 / 0.56 | 0.33 / 0.72 | 0.29 / 0.65 |
| GeoBit* [25] | 0.31 / 0.65 | 0.35 / 0.77 | 0.20 / 0.47 | 0.32 / 0.71 | 0.30 / 0.66 |
| Ours (DEAL) | 0.33 / **0.68** | 0.38 / **0.85** | 0.27 / 0.63 | **0.36** / **0.80** | 0.34 / **0.75** |

a) DaLI - 126 inliers  b) GeoBit - 179 inliers  c) DEAL - 208 inliers

Figure 3: **Matching example of deformed shirt.** Correct matches are drawn as green, and wrong ones as red lines. Our descriptor better handles the strong deformation (highlighted in the zoomed boxes), among the deformation-aware competitors.

affected by either illumination or viewpoint rigid changes (homography). Our descriptor presented a competitive performance in this dataset, even though it does not contain non-rigid deformations. These additional experiments can be checked in our supplementary material.

**Rotation and scale invariance.** To evaluate the robustness of our descriptor to rotation and scale, we select sequences with strong rotation and scale transformations, where the camera suffers in-plane rotations from $0°$ to $180°$ with $10°$ degrees steps for rotation tests. For the scale tests, the camera was moved backward in the Z direction, producing downscaling of the object. From Figure 4, one can notice that our method holds the best invariance to image in-plane rotations, and its very close to GeoBit in the scale sequences, although our method only requires monocular images.

## 4.3 Ablation studies and processing time

To evaluate the contribution of the components of our architecture and support our implementation decisions, we performed the ablation analysis of different parts of the proposed method.

**Contribution of the TPS warper.** We consider four different setups: (i) the use of the TPS on a standard regular grid patch, (ii) training the proposed architecture on a new dataset containing only rigid planar objects, (iii) using the second STN network $\Omega$ with fixed parameters

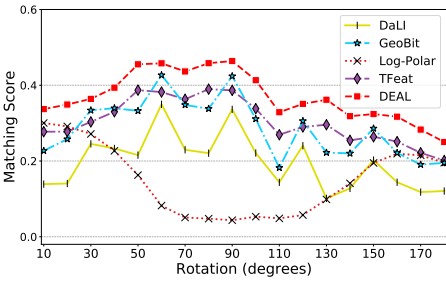 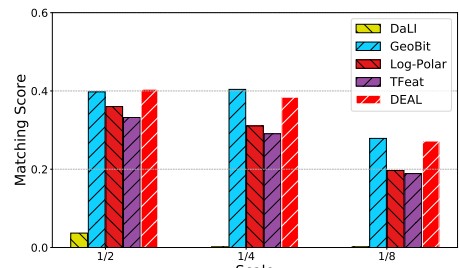

Figure 4: **Rotation and scale invariance.** Matching score curves obtained by matching SIFT keypoints showing results for rotation and scale for each target frame relative to the reference using the *Simulation* dataset. This experiment evaluates the robustness of the descriptors to both deformation, scale, rotation, and illumination in the most challenging sequence.

at the identity (Fixed STN polar sampler – akin to Log-Polar); and (iv) with the NRW component for modeling the deformations. In this analysis, we use the scale simulation sequences, that were purposefully chosen because they exhibit mostly strong deformations across multiple image pairs. The MMA achieved by each configuration can be seen in Table 3 (best in bold). All variations were trained until convergence with the same training procedure and HardNet initialization.

The ablation tests show that the polar sampler can improve matching accuracy compared to the Cartesian counterpart, and the non-rigid dataset also helps to improve accuracy. Most importantly, the NRW module alone can improve the accuracy of the descriptors by 4 p.p. when keypoints are affected by deformations, an important gain for tasks that require high matching accuracies such as deformable surface registration and non-rigid reconstruction.

Table 3: Ablation.

| Method | MMA $\uparrow$ |
|---|---|
| Cartesian NRW | 0.68 |
| Planar Data NRW | 0.75 |
| Fixed Polar STN | 0.75 |
| Polar NRW | **0.79** |

**Use of pre-trained ResNet features.** We also verified if transfer learning from a pre-trained ResNet model on ImageNet is feasible for the task of patch rectification. However, the training of the network did not converge well when using the pre-trained weights. We argue that the non-rigid warper module is focused on learning warping parameters, which is a considerably distinct task compared to classification. Thus, we initialize the weights of the entire network and train it from scratch.

**Processing time.** We executed the three deformation-invariant descriptors (DaLi, GeoBit, and DEAL) on a set of 250 descriptors (from $640 \times 480$ resolution images), running on a Intel (R) Core (TM) i7-7700 CPU @ 3.60 GHz and a GTX 1080 Ti GPU. Our descriptor was significantly faster than DaLI and GeoBit. While our method (GPU) spent 0.03 seconds to compute the descriptors, DaLI (CPU-only) and GeoBit (CPU-only) spent 112.95 and 33.72 seconds, respectively.

### 4.4 Applications

**Object Retrieval.** To further demonstrate the effectiveness of our descriptor in potential real-world applications, we performed experiments in two complementary real-world applications: object retrieval and tracking of deformable objects (more results can be found in the supplementary material).

In the retrieval application, we used a Bag-of-Visual-Words approach [8]. For each descriptor, we first construct a visual dictionary used to compute a global descriptor for each image. Given a query image, we compute the global descriptor and use $K$-Nearest Neighbor search to obtain the top $K$ closest objects. The metric

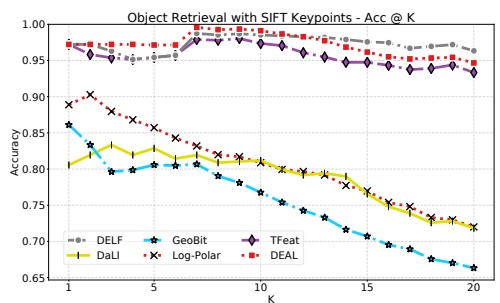

Figure 5: **Object retrieval.** Our descriptor outperforms all descriptors with $K \leq 10$, including the state-of-the-art retrieval descriptor DELF.

Table 4: **Tracking average LPIPS and RANSAC inliers rate.** Approaches with * are computed on their own detected keypoints, while ** uses RGB-D images. Best in bold and second-best underlined.

|  | DaLI | DELF* | GeoBit** | Log-Polar | TFeat | DEAL |
|---|---|---|---|---|---|---|
| Inliers RANSAC ↑ | 0.32 | 0.25 | **0.46** | 0.36 | 0.32 | **0.46** |
| LPIPS ↓ | 0.44 | 0.33 | **0.23** | 0.24 | 0.24 | 0.24 |

used is the retrieval accuracy, *i.e.*, the number of correct objects retrieved in the top $K$ images. Figure 5 shows the results with the best performing descriptors according to the matching experiments. Our method achieved the best accuracy with $K \leq 10$. In this experiment, we also consider DELF [39], the state-of-the-art descriptor designed and trained for image retrieval. Since DELF does not allow the description on commonly adopted keypoint detectors, we perform its evaluation using its own detected keypoints. The results indicate that our deformation-aware network was able to provide good results in a different but related task.

**Non-rigid tracking results.** This task consisted of tracking a region-of-interest of a template image over time using the correspondences from the computed descriptors. We approximate the deformation using TPS warping, which was combined with a RANSAC filtering to remove outlier correspondences for all descriptors, following the protocol proposed in [43].

Table 4 shows the average values of the Perceptual Patch Similarity distance (LPIPS) [49] and the number of inliers found by RANSAC for the main competitors. We also included DELF using their own detected keypoints on all sequences. As can be seen, even using RGB data only, our descriptor showed similar results to GeoBit. Representative qualitative results are shown in Figure 6 for the sequences *Shirt3* and *kanagawa_rot*. While the correspondences using our descriptor (last row) allowed the tracking of frames even in extreme conditions of rotation and deformation, DaLI was particularly affected by the object viewing changes.

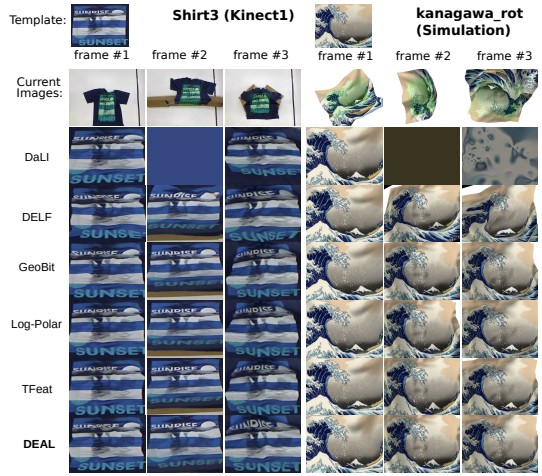

Figure 6: **Tracking results.** Qualitative tracking results for two sequences. Please notice the performance of the proposed approach (last row) for these challenging frames.

## 5 Conclusions

This paper presents a new local descriptor robust to non-rigid deformations, scale, and rotation transforms. The local features are extracted by a deformation-aware architecture trained in an end-to-end manner following a deformation guidance strategy. Besides providing highly discriminative and invariant feature vectors, our approach requires only still images and is time-efficient. We demonstrate the effectiveness and robustness of our descriptor by extensive experiments comprising the comparison with state-of-the-art descriptors, ablation studies, and its use on two real applications.

A limitation of our work is that its performance is compromised when dealing with surfaces having sharp deformations, discontinuities, reflections, and poor keypoint detector repeatability. These conditions also hamper the competitors. A more tailored discussion on the limitations of our method, including visual results and examples of failure cases of all methods can be checked in our supplementary material. Since matching keypoints across images is a fundamental step in many applications, our proposed method could be potentially used in surveillance contexts, for instance, recognizing and tracking humans from images. In this circumstance, the misuse of such technology could negatively affect people's privacy and mobility rights.

## Acknowledgments and Disclosure of Funding

The authors would like to thank CAPES (#88881.120236/2016-01), CNPq, FAPEMIG, and Petrobras for funding different parts of this work. R. Martins was also supported by the French National Research Agency through grants ANR MOBIDEEP (ANR-17-CE33-0011), ANR CLARA (ANR-18-CE33-0004) and by the French Conseil Régional de Bourgogne-Franche-Comté.

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
