# SUPPLEMENTARY MATERIAL:
# Extracting Deformation-Aware Local Features by Learning to Deform

**Guilherme Potje**[*]
Universidade Federal de Minas Gerais

**Renato Martins**
Université Bourgogne Franche-Comté

**Felipe Cadar**
Universidade Federal de Minas Gerais

**Erickson R. Nascimento**
Universidade Federal de Minas Gerais

In this supplementary material, we provide a detailed performance evaluation of the local descriptors using the same set of SIFT keypoints by expanding Table 2 from the main paper regarding the individual sequences of each dataset (please check Table 5), and an additional experiment on the dataset of planar scenes HPatches [1] (see Table 6). Furthermore, we also include additional qualitative results of the non-rigid object retrieval and tracking applications. To conclude, we provide details about the implementation, simulation framework, and submission checklist.

## A  Detailed Performance Evaluation with SIFT Keypoints

Table 5 shows the matching scores obtained by each descriptor for the individual sequences, allowing a more detailed performance assessment of the local descriptors. Our descriptor achieves the best performance in most sequences and the second-best performances in the remaining ones. The recently proposed GeoBit descriptor achieves the second-best results in most sequences. It is noteworthy that GeoBit uses both visual and depth information from RBG-D images, which contains accurate scene geometry. Our method, in contrast, only requires RGB images since our proposed deformation-aware network is trained to deform the input images to rectify image patches, making the descriptors robust to non-rigid deformations. Moreover, our method is remarkably faster than GeoBit and DaLI, which are the methods that take into account non-rigid deformations in the extraction process to compute local descriptors.

We also evaluate the impact of re-training the learning-based local descriptors TFeat [9] and Log-Polar [2] using our proposed dataset of simulated non-rigid deformations. They are shown as TFeat** and Log-Polar** in Table 5. We had to adapt minor parts of the original implementation of the authors. Still, with the best of our efforts, we tried to keep the implementation as close as possible to the original ones. We observed that TFeat** did not enjoy any benefit when re-trained in our dataset; however, Log-Polar** benefited from re-training. We hypothesize that TFeat did not improve since it uses a Cartesian sampling, which is more sensitive to local transformations in the patches. On the other hand, Log-Polar can take advantage of its robustness to local transformations due to the log-polar sampling to learn more stable features under deformations. We have re-trained the methods from scratch using their original initialization for the same 10 epochs we used to train our approach. In addition, all seeds for the sampling of image pairs and keypoints are kept the same as we did for our method.

---

[*]corresponding author's e-mail: `guipotje@dcc.ufmg.br`

35th Conference on Neural Information Processing Systems (NeurIPS 2021).

Table 5: **Detailed evaluation from reported results in Table 2 of the main manuscript for descriptors using the same set of SIFT keypoints**. Our descriptor is able to provide the highest matching scores in most sequences. Best in bold, second-best underlined, * denotes that the methods use RGB-D data, and ** indicates that the method was re-trained on our non-rigid dataset.

| DSet | Object (# pairs) | Avg. Matching Scores | | | | | | | | | | | |
| | | BRAND* | DAISY | DaLI | FREAK | GeoBit* | Log-Polar | Log-Polar** | ORB | SIFT | TFeat | TFeat* | DEAL |
|---|---|---|---|---|---|---|---|---|---|---|---|---|---|
| **Kinect 1** | Shirt2 (18) | 0.24 | 0.30 | 0.35 | 0.33 | **0.43** | 0.36 | 0.39 | 0.25 | 0.28 | 0.32 | 0.31 | 0.42 |
| | Shirt1 (14) | 0.17 | 0.26 | 0.25 | 0.26 | 0.31 | 0.30 | 0.31 | 0.20 | 0.23 | 0.26 | 0.25 | **0.34** |
| | Blanket1 (15) | 0.15 | 0.25 | 0.28 | 0.22 | 0.31 | 0.29 | 0.30 | 0.20 | 0.23 | 0.26 | 0.24 | **0.35** |
| | Can1 (6) | 0.07 | 0.05 | 0.09 | 0.07 | 0.20 | 0.16 | 0.13 | 0.05 | 0.06 | 0.13 | 0.11 | 0.20 |
| | Shirt3 (17) | 0.21 | 0.27 | 0.28 | 0.28 | 0.33 | 0.30 | 0.33 | 0.22 | 0.23 | 0.27 | 0.26 | **0.34** |
| **Kinect 2** | redflower_l (29) | 0.18 | 0.21 | 0.29 | 0.25 | 0.31 | 0.22 | 0.24 | 0.18 | 0.20 | 0.20 | 0.21 | 0.31 |
| | toucan_l (29) | 0.26 | 0.35 | 0.37 | 0.38 | 0.34 | 0.36 | 0.38 | 0.29 | 0.31 | 0.34 | 0.34 | **0.43** |
| | redflower_m (29) | 0.14 | 0.18 | 0.24 | 0.21 | 0.26 | 0.19 | 0.21 | 0.15 | 0.17 | 0.18 | 0.18 | **0.27** |
| | toucan_h (29) | 0.24 | 0.31 | 0.35 | 0.34 | 0.32 | 0.33 | 0.34 | 0.26 | 0.28 | 0.29 | 0.30 | **0.41** |
| | doramaar_l (29) | 0.28 | 0.35 | 0.41 | 0.41 | 0.40 | 0.35 | 0.37 | 0.34 | 0.34 | 0.35 | 0.35 | **0.43** |
| | toucan_m (29) | 0.26 | 0.33 | 0.38 | 0.36 | 0.34 | 0.32 | 0.34 | 0.28 | 0.29 | 0.30 | 0.31 | **0.41** |
| | lascaux_l (29) | 0.33 | 0.46 | 0.52 | 0.53 | 0.56 | 0.48 | 0.50 | 0.44 | 0.45 | 0.47 | 0.48 | **0.58** |
| | redflower_h (29) | 0.12 | 0.12 | 0.17 | 0.16 | 0.18 | 0.13 | 0.15 | 0.11 | 0.11 | 0.12 | 0.13 | **0.19** |
| | blueflower_h (29) | 0.20 | 0.28 | 0.37 | 0.30 | 0.34 | 0.30 | 0.31 | 0.24 | 0.25 | 0.27 | 0.26 | **0.39** |
| | blueflower_m (29) | 0.19 | 0.27 | 0.35 | 0.30 | 0.36 | 0.30 | 0.31 | 0.23 | 0.24 | 0.26 | 0.26 | **0.38** |
| | blueflower_l (29) | 0.27 | 0.29 | 0.37 | 0.36 | 0.36 | 0.30 | 0.32 | 0.24 | 0.24 | 0.27 | 0.27 | **0.40** |
| **DeSurT** | campus (29) | 0.16 | 0.22 | 0.30 | 0.17 | 0.23 | 0.28 | 0.29 | 0.21 | 0.22 | 0.25 | 0.25 | **0.33** |
| | sunset (29) | 0.08 | 0.12 | 0.11 | 0.13 | 0.12 | 0.15 | 0.15 | 0.09 | 0.11 | 0.12 | 0.12 | 0.15 |
| | scene (29) | 0.21 | 0.19 | 0.29 | 0.20 | 0.23 | 0.28 | 0.30 | 0.23 | 0.24 | 0.27 | 0.27 | **0.33** |
| | newspaper1 (29) | 0.16 | 0.23 | 0.27 | 0.19 | 0.32 | 0.32 | 0.33 | 0.24 | 0.26 | 0.28 | 0.25 | **0.36** |
| | cobble (29) | 0.16 | 0.09 | 0.21 | 0.15 | 0.21 | 0.24 | 0.26 | 0.21 | 0.22 | 0.22 | 0.22 | **0.31** |
| | cushion1 (29) | 0.11 | 0.17 | 0.15 | 0.14 | 0.16 | 0.20 | 0.22 | 0.13 | 0.14 | 0.18 | 0.16 | **0.23** |
| | brick (29) | 0.20 | 0.16 | 0.31 | 0.22 | 0.26 | 0.28 | 0.30 | 0.25 | 0.25 | 0.26 | 0.26 | **0.34** |
| | cushion2 (29) | 0.07 | 0.06 | 0.07 | 0.08 | 0.09 | 0.09 | 0.08 | 0.06 | 0.06 | 0.07 | 0.06 | **0.11** |
| **Simulation** | lascaux_scale (3) | 0.02 | 0.10 | 0.02 | 0.08 | **0.43** | 0.35 | 0.36 | 0.06 | 0.33 | 0.33 | 0.31 | 0.42 |
| | chambre_scale (3) | 0.02 | 0.06 | 0.01 | 0.03 | 0.22 | 0.17 | 0.19 | 0.03 | 0.14 | 0.16 | 0.15 | 0.22 |
| | kanagawa_scale (3) | 0.02 | 0.11 | 0.01 | 0.06 | 0.42 | 0.35 | 0.39 | 0.04 | 0.30 | 0.33 | 0.34 | 0.42 |
| | lascaux_rot (18) | 0.08 | 0.36 | 0.26 | 0.31 | 0.37 | 0.18 | 0.42 | 0.33 | 0.37 | 0.38 | 0.38 | **0.46** |
| | chambre_rot (18) | 0.06 | 0.25 | 0.20 | 0.18 | 0.24 | 0.15 | 0.29 | 0.20 | 0.24 | 0.26 | 0.25 | **0.32** |
| | kanagawa_rot (18) | 0.06 | 0.22 | 0.13 | 0.21 | 0.25 | 0.12 | 0.31 | 0.19 | 0.25 | 0.27 | 0.26 | **0.33** |
| Mean | * | 0.16 | 0.22 | 0.25 | 0.23 | 0.30 | 0.26 | 0.29 | 0.20 | 0.23 | 0.26 | 0.25 | **0.34** |

# B  Performance Evaluation on HPatches

We selected the widely adopted HPatches [1] dataset containing predominantly rigid objects but under severe viewpoint and illumination changes. The dataset is composed of images from 116 planar scenes with 5 pairs of images each (we selected the "Full image sequences"). The dataset is split between pairs affected either by viewpoint or illumination changes, and the results are presented independently for each split (following the protocol presented in Log-Polar [2], we considered 51 scenes in the illumination split or 50 in the viewpoint split). We detected $1,000$ SIFT keypoints independently for each image, which are used for all descriptors except for ASLFeat [4], LF-Net [7] and R2D2 [8], which detect their own $1,000$ keypoints. For the evaluation, we adopted the matching score (MS) metric, as explained in the main paper. The illumination and viewpoint splits have 255 and 250 pairs of images, respectively.

The performance of our method for matching planar scenes can be seen in Table 6. Results demonstrate that the proposed descriptor capabilities display competitive results on images without deformed objects using the keypoints detected by SIFT. Our method also works well with ASLFeat keypoints (ASL+DEAL). To run our method with ASL keypoints, we have empirically tested several scales to be used with our descriptor and chose the value of 2.5 for all keypoints (no keypoint orientation is required for DEAL).

Table 6 presents the matching scores in the illumination and viewpoint splits, as well as the average; our method is close to the recent state-of-the-art descriptors, specifically designed and trained for matching rigid objects (images affected by homography). Those methods were also trained with real images. Worth mentioning is the fact that R2D2 and ASLFeat are not invariant to image in-plane

Table 6: **Evaluation on HPatches dataset.** Although HPatches does not contain non-rigid deformations (which penalizes our method), our descriptor exhibit competitive results among the state-of-the-art descriptors.

| Rank | Desc | MS view. ↑ | MS illu. ↑ | MS avg. ↑ |
|------|------|-----------|-----------|-----------|
| 1 | ASLFeat | 0.34 | 0.44 | 0.39 |
| 2 | ASL+DEAL | 0.33 | 0.35 | 0.34 |
| 3 | R2D2 | 0.26 | 0.38 | 0.32 |
| 4 | DEAL | 0.26 | 0.23 | 0.25 |
| 5 | TFeat | 0.26 | 0.24 | 0.25 |
| 6 | SIFT | 0.24 | 0.23 | 0.23 |
| 7 | LF-Net | 0.20 | 0.23 | 0.22 |
| 8 | Log-Polar | 0.09 | 0.29 | 0.19 |

rotations; thus, in this dataset, our descriptor is doubly penalized by its invariance to transformations that do not exist in HPatches. Nevertheless, as demonstrated, our descriptor achieves competitive performance among the descriptors using the same set of keypoints (TFeat, Log-Polar, SIFT), which are specifically designed to work with planar scenes and were trained on real-world images.

## C  Additional Qualitative Results

We also provide additional qualitative visualizations from the results shown in Table 4 and Figure 5 of the paper, as well as a more detailed result of the object retrieval experiments.

**Content-Based Object Retrieval.**  For the object retrieval, we used the union of the *Kinect2*, *DeSurt*, *Kinect1*, and *Simulation* datasets to compose a database of 24 different objects. Each reference frame containing the undeformed object is defined as the set of queries. The remaining images of the datasets are used as a retrieval database. For each method, we computed descriptors from a large set of keypoints extracted from the retrieval database. We randomly sampled 10,000 descriptors to construct a visual dictionary, using the K-medoid method with $k$ set to 10. We employ the K-medoids method to assure compatibility between binary and floating-point descriptors. All descriptors, except for DELF, were extracted using the same SIFT keypoints. We then extracted a normalized frequency histogram for each image using one bin for each visual word from the constructed dictionary. To retrieve the objects of a query, we compute the normalized frequency histogram of the query and use k-nearest neighbor search to retrieve the top $K$ objects on the database closest to the query. The metric used is the accuracy over the top $K$ retrieved objects ranked by their Euclidean distance from the query, $K = \{1, 2, \ldots, 20\}$. Qualitative results can be observed in Figure 7. Our method can retrieve heavily deformed objects under challenging image transformations, while other methods fail to recover correct objects over their top results.

**Non-Rigid Object Tracking.**  We also present additional tracking qualitative results from sequences affected by moderate to extreme deformations. We report tracking failure situations where the resulting brute force association from the descriptors did not provide enough inliers correspondences for the RANSAC outlier rejection procedure to filter spurious matches. In this case, even a few outliers can completely deteriorate the thin-plate-spline warp estimations. For all sequences and all methods, we set RANSAC iterations to 1,500. Some extreme and ambiguous feature description cases are shown in Figure 8 for the sequences *Can1* and *cushion2*. In these sequences, the objects are affected by extreme deformations and reflections. Note that even the best methods fail in such cases.

A limiting factor particular to our approach, is when extreme deformations happen. In this case, even if the non-rigid warper module could correctly find the warping transform and correctly rectify the input image, there would be a loss of data due to the interpolation of sub-sampled pixels from the image.

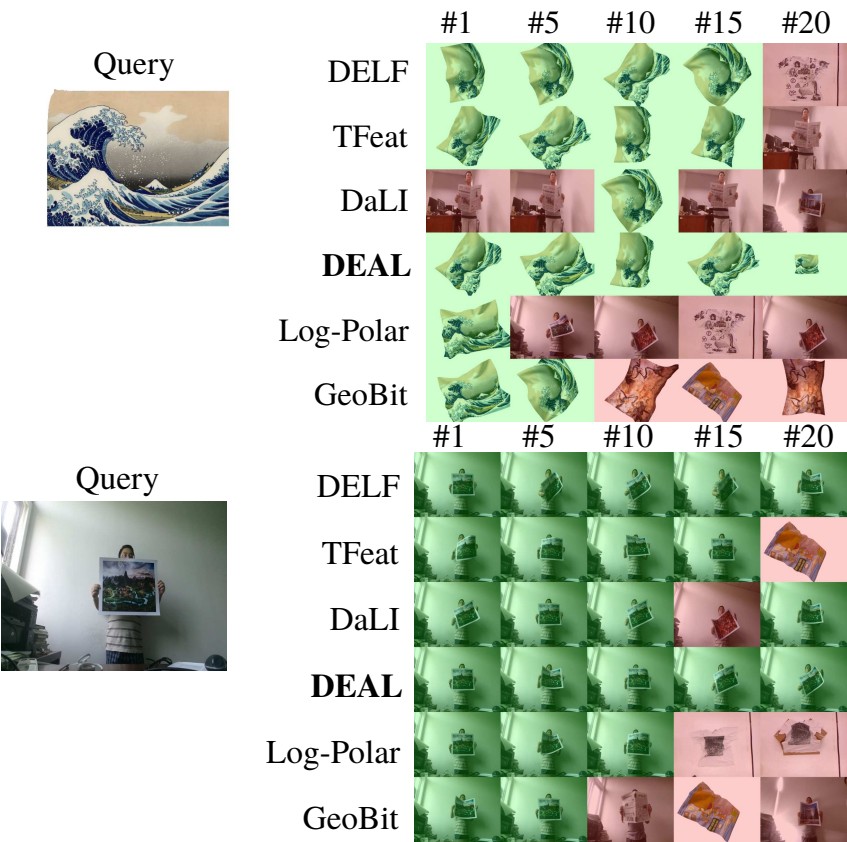

Figure 7: **Qualitative results for object retrieval.** *Top:* Results for Kanagawa sequence (Simulation dataset); *Bottom*: Scene sequence (real-world data from DeSurT dataset).

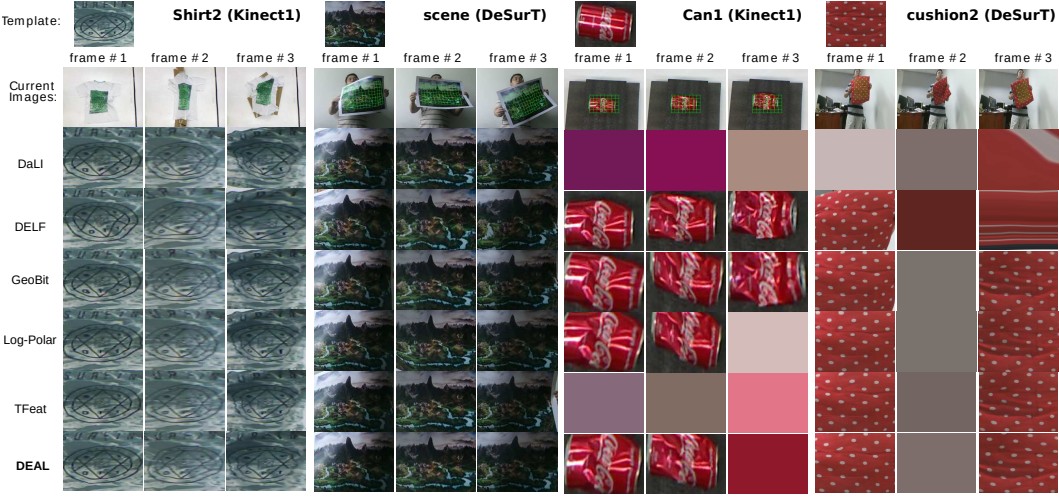

Figure 8: **Qualitative results for the tracking task.** Qualitative tracking results on four sequences with different difficulty levels: *Shirt2* and *scene* with moderate deformations, rotations and view changes, and; *Can1* and *cushion2* affected by extreme deformations.

# D   Simulation Framework Details

Our physics simulation framework is implemented in OpenGL, achieving the necessary efficiency and flexibility to allow low-cost generation of thousands of images of realistic deforming objects

suitable for training Deep Neural Networks, permitting efficient low-level access to the simulation data, i.e., Z-buffer, camera parameters, and perfect correspondence in the sequences. The objects are represented by a grid of particles having mass in 3D space. Deformations are induced by the forces applied onto them, implemented as wind and gravity. A constraint satisfaction optimization step is performed over all particles to enforce a constant distance of neighboring particles, thus keeping the deformation isometric. For each simulation round, we generate (i) random wind forces in all directions to generate diverse object deformations in a chaotic fashion, (ii) illumination variation such as intensity, global position, number of light sources, directional lighting, and color changes to enforce realistic non-linear illumination diversity, and (iii) Gaussian noise in image pixels to simulate real camera sensors.

## E    Additional Implementation Details

We provide a detailed linear list of steps from the input to the output of the entire model of Figure 9 for an input image with a single keypoint for the sake of clarity. Assume a tensor shape convention of $(H = Height, W = Width, C = Channels)$. Let the input of the network be a $(800, 800, 3)$ image with a single keypoint having parameters $(x = 80, y = 80, size = 12.0)$ exactly at a deforming surface:

1. The whole input image is forwarded to the ResNet-34 block, giving an output tensor $\mathbf{X}$ of shape $(100, 100, 128)$.

2. The first STN $\Psi$ takes the keypoint parameters $(x = 80, y = 80)$ of the keypoint, and rescales those coordinates by $1/8$, since these coordinates are in the original image and $\mathbf{X}$ is spatially downscaled by $1/8$. Notice that the keypoint size is not considered in the first STN. The first STN samples a local patch of features of $\mathbf{X}$ with an affine matrix (the matrix encodes a translation of the keypoint position). The first STN then converts a $5 \times 5$ identity mesh grid $\mathcal{M}_I$ (a mesh grid of $5 \times 5$ pixel coordinates centered at the origin) by translating it to become the transformed grid $\mathcal{M}_\theta$, a mesh grid centered at $(10, 10)$, i.e., at the position of the keypoint, by using the affine matrix. Finally, the Sampler Layer samples a local tensor $(5, 5, 128)$ from $\mathbf{X}$ at the locations from $\mathcal{M}_\theta$. The resampling is performed for each keypoint; thus, each keypoint has a different local tensor if they lie in

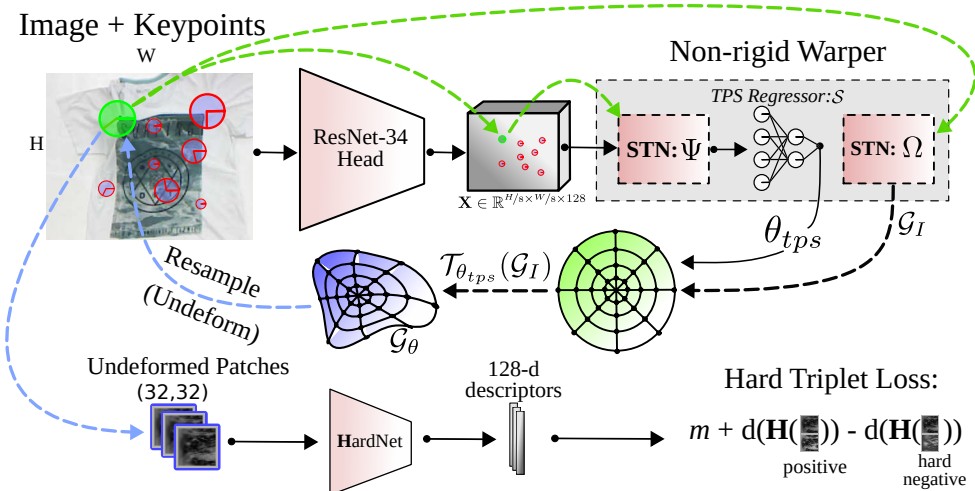

Figure 9: **Proposed formulation for computing descriptors of deforming objects.** The non-rigid warper component undeforms local image patches by applying a learned deformation for each keypoint. Higher level image information such as appearance and shape are encoded globally by the ResNet feature tensor $\mathbf{X}$. Then, the two carefully designed spatial transformers and the TPS regressor network decode the information locally, by estimating the warp parameters $\theta_{tps}$ used to rectify patches from the original image. The green arrows imply keypoint attribute information. The rectified patches are fed to HardNet that extracts discriminant descriptors. The full network model is trained end-to-end by optimizing the hard triplet loss.

different spatial coordinates in the image. As we have only one keypoint in this example, there will be only one local tensor of shape $(5, 5, 128)$. Please notice that in this step no deformation has been applied. We just resampled a local tensor from the ResNet-34 features centered at the position of the keypoint using bilinear interpolation. This local resampling is analogous to cropping a patch of size $5 \times 5$ pixels in the original input image as done by classic hand-crafted local descriptors in Computer Vision such as SIFT or learned-based as TFeat and Log-Polar. However, in our network we use higher-level features coming from the ResNet feature maps, and differentiable sampling to perform the cropping task. Our experiments show evidence that instead of directly sampling the input image, the ResNet-34 is able to reason about higher-level contextual cues about the object, such as its shape, the scene illumination, and the relative configuration of the local texture of the object, to find the right local deformation parameters.

3. This is where the second STN $\Omega$ begins. Both the Polar and TPS transformations are inside this STN block. The TPS regressor network $\mathcal{S}$ takes as input the flattened version of the "cropped" tensor from the previous layer $(5 * 5 * 128)$ and generates as output $\theta_{tps}$, which is a parameter vector used by Equation 2.

4. The second Spatial Transformer module $\Omega$ first generates an identity grid $\mathcal{G}_I$ using the keypoint attributes $(x = 80, y = 80, size = 12.0)$, and *only depends on the keypoint attributes*. The identity grid is created in the original image spatial coordinate space, therefore, for our single keypoint, the identity polar grid will be centered at coordinates $(80, 80)$ with radius of 12 pixels on the input image.

5. Using the TPS parameters $\theta_{tps}$ and $\mathcal{G}_{\mathcal{I}}$ obtained from $\Omega$, we apply Equation 2 of the paper to warp $\mathcal{G}_{\mathcal{I}}$ into the deformed grid $\mathcal{G}_\theta$, which is then used by the Sampler Layer to sample pixels in the original image. The result is the rectified patch, which is expected to be invariant to local deformations. The invariance to deformations is enforced by the triplet loss in the training phase, since the network is fully differentiable and the NRW module learns meaningful deformations that increases the distinctiveness of the final descriptor.

6. At last, the rectified patch is forwarded to the HardNet that outputs a 128-D descriptor that will be robust to non-rigid deformations. In summary, if the network receives two different images with the single keypoint centered on a deforming surface, the ResNet features will be different when the local shape, relative texture, and illumination change. As a result, for the same keypoint, $\theta_{tps}$ will be different for each input image. Therefore, the difference between the ResNet features extracted from the two input images will encode the relative local transformations that the TPS regressor needs, in order to sample the two patches in both defomed surfaces.