# OpenReview forum: "Extracting Deformation-Aware Local Features by Learning to Deform"
_NeurIPS.cc/2021/Conference — NeurIPS 2021 Poster_

### Official Review · Reviewer_P78p · 2021-06-30

**Rating:** 7
**Confidence:** 3

**Summary:**

The paper proposes DEAL, an RGB patch descriptor that has a learned robustness against non-rigid deformations. Given a SIFT keypoint, surrounding ResNet features are used to estimate a non-rigid thin-plate spline deformation of the patch. The RGB patch, after being rectified by the thin-plate spline, is passed to a CNN that regresses a 128-dim descriptor. The network is trained via a hard triplet loss. The training data is simulated and uses groundtruth correspondences/matches, but results are shown on both real and synthetic data. Experiments compare to a number of classic and learned baselines. DEAL outperforms other methods in the main experiment and shows best or second-best performance on a couple of applications.

**Limitations And Societal Impact:**

Limitations are mentioned in one sentence (lines 371-372) but not discussed to give a good sense of them, and I am unable to locate the limitations that are supposed to be in the appendix. Qualitative results could be added.

Potential negative societal impact is not discussed. Better tracking of clothed people (which have non-rigid deformations) through surveillance might be one point.

**Main Review:**

The proposed method is a novel descriptor that tackles non-rigid deformations, which is a very desirable goal. The experiments on real data show promising results, especially considering that the descriptor was trained on only synthetic data. My main concern is a better attribution of where the improvement comes from: data or architecture (thin plate splines)? I find the current experiments unsatisfying in this regard, the next paragraph goes into details.

The "Contribution of the TPS warper" ablation study is interesting. My understanding is that "(i)"/PTN is a pipeline that is the same (up to maybe number of conv layers etc.) as Fig. 3 in Ebel et al.'s Log-Polar descriptor [12]: sample the original image with a circular polar grid around the keypoint and then feed that patch through a CNN to get the descriptor. Table 2 shows an MMA of 0.49 for Log-Polar on the Simulation dataset, while DEAL has 0.80. The ablation study shows 0.79 for DEAL (i.e. with NRW) and 0.75 for the PTN ablation. The ablation uses a specific scene from the Simulation dataset, which explains the differing performance of DEAL. I interpret the discrepancy between PTN ablation (0.75) and Log-Polar (0.49) to mean that most of the improvement that DEAL shows over Log-Polar is due to training data (from 0.49 to 0.75) rather than architecture (from 0.75 to 0.79). Is that correct? If so, I believe that further baseline experiments for Table 2 would be necessary that train the two learning baselines (TFeat and Log-Polar, lines 218-219) with the proposed *non-rigid* simulated dataset (lines 229-239) instead of their original datasets, which seem to be *rigid*.

I have a few bigger questions about the paper, but which I do not expect to sway the acceptance decision:

- Limitations should be discussed more, including qualitative examples.
- What is the intuition behind using the first spatial transformer? What would happen if it isn't used and Y = X? Such an ablation experiment should be added.
- The sensibility analysis in Table 1 shows that increasing the margin to 0.75 and increasing the STN output to 5 x 5 improves over the baseline. Why is only the latter change adapted (lines 266-267)? The margin change gives 50% as much improvement as changing the STN output does after all.
- The STN is meant to normalize out some spatial transforms around the keypoint. But how can the localization network then determine the right (desirable) rectifying transformation for the original patch around the keypoint if the localization network doesn't have access to the uncorrected spatial layout of the original patch? I.e. the STN removes some information about the spatial layout of the patch, then the TPS transform is estimated from that reduced information, but then the TPS tranform is applied to the original patch. Does such a setup make sense?
- Is there a reason that the DeepDeform dataset [4] was not used for evaluation? Is it not suited for the task?

There are a number of smaller clarifications that I suggest a revised version should take into account:

- What is the size of the receptive field of the ResNet features after the sixth convolutional block (line 134)?
- In lines 156-158, it would be good to mention shortly that the parameters theta get *regressed* by the spatial transformer from X. It is not fully clear whether theta is the same globally for all H/s x W/s pixels or whether it differs between them.
- The MLP from line 178 is not convolutional, so are the 5 x 5 128-dim feature vectors stacked before passing them to the MLP?
- Does T_theta in line 159 refer to the STN in Fig. 2? If so, the notation for T_theta(G_I) in Fig. 2 should be changed to avoid confusion.
- It would also be good to add the missing notations to Fig. 2, e.g. the TPS parameter regressor S (line 152), M_theta and M_I (line 159), the warped features Y (line 161).
- What does the "TPS" in the green box in Fig. 2 stand for? Doesn't the TPS regressor S regress theta_tps directly?
- I don't understand the sentence in lines 186-188. Why is the regular (not the TPS-transformed) polar grid applied to the image (not just the patch)? What are the keypoint's attributes, where do they come from, have they been mentioned before? Why are they needed to get an identity transform? What is the difference between scale and size?
- The sentence in lines 196-198 is a bit ambiguous and could be split into two shorter sentences, and the sentence in lines 198-199 could be pulled in front. "which" could refer to the loss or to HardNet. It is not obvious that HardNet comes from [22], only that the triplet loss is used from [22].
- A short explanation of what needed to change for adapting to R2D2 and ASLFeat (lines 282-283) in the appendix would be good for completeness.

Update after rebuttal ---

I initially had a similar difficulty in understanding the usages of two STNs as reviewer fuQg, which made me question the rationale behind the method design and potential issues with it. Given the authors' clarifications, I corrected my understanding (the first STN uses a regular, straight grid) and am now satisfied with the design. Similarly, I was concerned about baselines (maybe training data is the crucial difference), which the new experiments in the rebuttal show to not be the case. Reviewer dECi brought up some more important ablations, where I find the results provided in the authors' response convincing. I find the remaining points mentioned by the reviewers to be addressed well by the authors' response.

However, there is one part that is not fully convincing me, namely comparison to existing work, as mentioned by reviewer dECi. The lack of non-rigid datasets in the literature suitable for evaluation is unfortunate, but the authors make an effort to introduce their own dataset and to evaluate on the couple of datasets that are suitable. The performance on rigid datasets is competitive, but not state-of-the-art, which I find good enough, given that the submission has to relax assumptions to be able to handle non-rigid deformations.

I will change my rating from 5 to 7.

**Time Spent Reviewing:**

4

---

> ### Author Response · Authors · 2021-08-10
> **Point-by-point reply to Reviewer P78p - Part I**
>
> _The proposed method is a novel descriptor that tackles non-rigid deformations, which is a very desirable goal. The experiments on real data show promising results, especially considering that the descriptor was trained on only synthetic data._
>
> Thanks for the positive comments about our work.
>
> _***P78p Q1:*** My main concern is a better attribution of where the improvement comes from: data or architecture (thin plate splines)? I find the current experiments unsatisfying in this regard, the next paragraph goes into details._
>
> ***P78p A1:*** The reviewer raised this important point, and we have further discussed and justified our contribution in the new results and answers below. In order to support our discussion, as kindly suggested by the reviews, we have performed new experiments to show the contributions of our designed model and data. The results and discussion are presented in the following answers.
>
> _***P78p Q2:*** The "Contribution of the TPS warper" ablation study is interesting. My understanding is that "(i)"/PTN is a pipeline that is the same (up to maybe number of conv layers etc.) as Fig. 3 in Ebel et al.'s Log-Polar descriptor [12]: sample the original image with a circular polar grid around the keypoint and then feed that patch through a CNN to get the descriptor. Table 2 shows an MMA of 0.49 for Log-Polar on the Simulation dataset, while DEAL has 0.80. The ablation study shows 0.79 for DEAL (i.e. with NRW) and 0.75 for the PTN ablation. The ablation uses a specific scene from the Simulation dataset, which explains the differing performance of DEAL. I interpret the discrepancy between PTN ablation (0.75) and Log-Polar (0.49) to mean that most of the improvement that DEAL shows over Log-Polar is due to training data (from 0.49 to 0.75) rather than architecture (from 0.75 to 0.79). Is that correct? If so, I believe that further baseline experiments for Table 2 would be necessary that train the two learning baselines (TFeat and Log-Polar, lines 218-219) with the proposed non-rigid simulated dataset (lines 229-239) instead of their original datasets, which seem to be rigid._
>
> ***Table: Comparison with state-of-the-art descriptors.*** The methods marked with * are computed on RGB-D images, and methods with ** were re-trained using our proposed non-rigid dataset. Results indicate that our proposed descriptor achieves state-of-the-art performance for matching deformable surfaces overall. In addition, our proposed dataset can also improve the robustness of existing methods to non-rigid deformations. Results are sorted by increasing MMA.
>
> |Method|Kinect1|Kinect2Sampled|DeSurTSampled|SimulationICCV| Avg. MS / MMA ↑|
> |:-----|-----:|-----:|-----:|-----:|-----:|
> |BRAND*|0.17 / 0.34|0.22 / 0.49|0.14 / 0.33|0.04 / 0.09|0.16 / 0.34|
> |R2D2|0.17 / 0.36|0.25 / 0.59|0.14 / 0.32|0.06 / 0.16|0.17 / 0.39|
> |ORB|0.19 / 0.39|0.25 / 0.55|0.18 / 0.40|0.14 / 0.30|0.20 / 0.43|
> |SOSNet|0.17 / 0.34|0.25 / 0.55|0.17 / 0.38|0.26 / 0.57|0.22 / 0.47|
> |DAISY|0.23 / 0.47|0.29 / 0.62|0.16 / 0.37|0.19 / 0.39|0.22 / 0.48|
> |FREAK|0.24 / 0.49|0.33 / 0.72|0.16 / 0.38|0.15 / 0.31|0.23 / 0.51|
> |SIFT|0.20 / 0.42|0.26 / 0.57|0.19 / 0.43|0.27 / 0.59|0.23 / 0.51|
> |DaLI|0.25 / 0.51|0.35 / 0.76|0.21 / 0.48|0.10 / 0.22|0.25 / 0.54|
> |TFeat**|0.23 / 0.48|0.28 / 0.62|0.20 / 0.46|0.28 / 0.61|0.25 / 0.55|
> |TFeat|0.25 / 0.50|0.28 / 0.61|0.21 / 0.48|0.29 / 0.63|0.26 / 0.56|
> |ASLFeat|0.31 / 0.58|0.39 / 0.69|0.28 / 0.53|0.19 / 0.35|0.31 / 0.56|
> |Log-Polar|0.28 / 0.58|0.30 / 0.65|0.23 / 0.54|0.22 / 0.49|0.26 / 0.57|
> |D2-Net|0.20 /0.50 |0.23 / 0.82| 0.14 / 0.47|0.11 / 0.30|0.17 / 0.57|
> |LFNet|0.44 / 0.40|0.51 / 0.43|0.28 / 0.77|0.21 / 0.74|0.36 / 0.59|
> |LIFT|0.09 / 0.57|0.16 / 0.65|0.08 / 0.52|0.13 / 0.73|0.12 / 0.62|
> |LogPolar**|0.29 / 0.60|0.31 / 0.69|0.24 / 0.56|0.33 / 0.72|0.29 / 0.65|
> |GeoBit*|0.31 / 0.65|0.35 / 0.77|0.20 / 0.47|0.32 / 0.71|0.30 / 0.66|
> |Ours|0.33 / 0.68|0.38 / 0.85|0.27 / 0.63|0.36 / 0.80|0.34 / 0.75|
>
> ***Table: Ablation.***
>
> | Method | MMA ↑|
> |:---------|-------:|
> |Cartesian NRW| 0.68|
> |Planar Data NRW| 0.75|
> |PTN-only| 0.75
> |Polar NRW| 0.79 |
>
>
> ***P78p A2:*** We thank the reviewer for the question and comments. We would like to clarify that one key aspect of our descriptor is the capability to leverage global information of the deformation affecting the image in the description. This key feature is notably relevant in the context of strong deformations, such as in the sequence used in the ablation experiment of Table 3, which is now improved by two new ablation tests as suggested by the reviewer (see the updated Table: Ablation). The new ablation tests show that (i) the PTN sampler is able to improve matching accuracy, (ii) the non-rigid dataset also helps to improve accuracy, and finally (iii) the NRW component also boosts performance. To elucidate the contribution of our proposed network architecture in the description of deformable surfaces, apart from the training data, we have re-trained TFeat and Log-Polar descriptors using our database. These results are shown in Table: Comparison with state-of-the-art descriptors, which is an updated version of Table 2 of the manuscript. This new experiment shows evidence that both the non-rigid dataset and our proposed architecture are able to improve the matching of deformable surfaces. The re-trained TFeat** on the non-rigid database did not improve the MS, and slightly decreased the MMA. On the other hand, our dataset increased the performance of the retrained Log-Polar from 0.57 to 0.64 MMA (7 p.p difference), but it is still under the performance of our descriptor scores. With the contribution of the designed full deformation-aware architecture, we achieved an additional boost of 11 p.p. in matching accuracy.
>
> _I have a few bigger questions about the paper, but which I do not expect to sway the acceptance decision:_
>
> _***P78p Q3:*** Limitations should be discussed more, including qualitative examples._
>
> ***P78p A3:*** We have extended the discussion of limitations and failure cases in the main paper, as well pointing out to the visual examples provided in the supplementary material. A limiting factor is the presence of extreme deformations. In this case, even if our networks could correctly find the warping transform and correctly rectify the input image, there would be a loss of data due to the interpolation of sub-sampled pixels from the image. We have added in the supplementary material challenging sequences with objects producing failure cases. We highlighted and added the discussion in a more explicit form in the paper to clarify this point.
>
> _***P78p Q4:*** What is the intuition behind using the first spatial transformer? What would happen if it isn't used and Y = X? Such an ablation experiment should be added._
>
> ***P78p A4:*** The first transformer is used to sample the ResNet-34 mid-level feature maps to be used by the deformation module. If one uses the identity (without regressing any TPS parameter), the transformation is not applied, and the sampling strategy would be equivalent to a polar sampling of the patch centered at the keypoint coordinates. This ablation study can be seen in Table 3 of the paper (PTN). We highlight that this is essential to capture the non-rigid deformations as shown in the qualitative examples and in the new results of Table: Comparison with state-of-the-art descriptors, where we included TFeat and Log-Polar re-trained in our non-rigid datasets.
>
> ***P78p Q5:*** _The sensibility analysis in Table 1 shows that increasing the margin to 0.75 and increasing the STN output to 5 x 5 improves over the baseline. Why is only the latter change adapted (lines 266-267)? The margin change gives 50% as much improvement as changing the STN output does after all._
>
> ***P78p A5:*** We thank the reviewer for the keen observation. We also tested both changes simultaneously while training the models. However, conversely to the situation observed of the improvement made by individually changing the parameters, applying both changes has slightly reduced the model's performance. Therefore we did not have applied both changes used in the experiments. We have added an additional paragraph discussing this point in the ablation analysis.
>
> _***P78p Q6:*** The STN is meant to normalize out some spatial transforms around the keypoint. But how can the localization network then determine the right (desirable) rectifying transformation for the original patch around the keypoint if the localization network doesn't have access to the uncorrected spatial layout of the original patch? I.e. the STN removes some information about the spatial layout of the patch, then the TPS transform is estimated from that reduced information, but then the TPS tranform is applied to the original patch. Does such a setup make sense?_
>
> ***P78p A6:*** We clarify to the reviewer that the STN is only used to re-sample local ResNet-34 feature tensors centralized at the keypoint coordinates in the image, which will be used by the regressor network as features. Those features encode contextual cues about the image, and serve as a guidance to the TPS regressor, which is enforced to learn meaningful deformations during training. We believe the spatial layout that the reviewer is referring to is the identity polar grid  $\mathcal{G}_{I}$, which is used as control points for the TPS warper. The TPS warper will receive the learned $\theta$ parameters and the identity grid to warp the identity grid into the goal coordinates $\mathcal{G}$ $_\theta$ . Then, $\mathcal{G}$ $_\theta$ is used to re-sample image pixels using bilinear interpolation to finally generate the rectified image patch.

---

> ### Author Response · Authors · 2021-08-10
> **Point-by-point reply to Reviewer P78p - Part II**
>
> ***P78p Q7:*** _Is there a reason that the DeepDeform dataset [4] was not used for evaluation? Is it not suited for the task?_
>
> ***P78p A7:*** We indeed have considered evaluating our approach also in the DeepDeform dataset. Notably because of the number of scenes provided and since acquiring accurate pixel-wise non-rigid object correspondences is cumbersome. Unfortunately, while inspecting their dataset, we first noticed the annotated sparse correspondences that we planned to use in the evaluation were often done with coarse accuracy (which justifies the suggested threshold of 20 pixels in their evaluation). The annotations were often located surprisingly in object regions without texture information, and they are also always repeatable. This again means that all keypoints are always visible, which is not the case in practical scenarios (notably under non-rigid deformations). Therefore, due to these characteristics with their annotated sparse keypoints, we chose to perform the evaluation in the two provided datasets with independently detected keypoints for each frame.
>
> _There are a number of smaller clarifications that I suggest a revised version should take into account:_
>
> _***P78p Q8:*** What is the size of the receptive field of the ResNet features after the sixth convolutional block (line 134)?_
>
> ***P78p A8:*** The size of the receptive field at the sixth convolutional block is a 3x3 convolution over the last feature map. We will include this information in the paper.
>
> _***P78p Q9:*** In lines 156-158, it would be good to mention shortly that the parameters theta get regressed by the spatial transformer from X. It is not fully clear whether theta is the same globally for all H/s x W/s pixels or whether it differs between them._
>
> ***P78p A9:*** Thank you for pointing this issue out, we will clarify this in the manuscript. The parameter theta is different for each keypoint.  First, for each keypoint, a local tensor feature map of size $5 \times 5$ is interpolated by the first STN using the coordinates of the keypoint in the image (downscaled to ⅛). This can be seen on the diagram of Fig. 2 in the paper (small red dots in the full tensor $\mathbf{X}$. For example, if there exists a keypoint at position (x = 80, y = 80) in the input image, the first STN would re-sample a $5 \times 5$ local tensor of shape (5,5,128) centered at position (x = 10, y = 10) of the feature map coming from ResNet, and then this feature map feeds the regressor network to estimate $\theta$.
>
> _***P78p Q10:*** The MLP from line 178 is not convolutional, so are the 5 x 5 128-dim feature vectors stacked before passing them to the MLP?_
>
> ***P78p A10:*** Yes, they are flattened to a 1D array before feeding them to the MLP.
>
> _***P78p Q11:*** Does T_theta in line 159 refer to the STN in Fig. 2? If so, the notation for T_theta(G_I) in Fig. 2 should be changed to avoid confusion._
>
> ***P78p A11:*** No, the $T_\theta$ is the the transformation that applies the deformation to the input grid (see Equation 2 in the paper), given the $\theta_{tps}$ and the input control points $\mathcal{G}_{I}$,which is the identity polar grid initialized with the keypoint size and the keypoint position in the original image.
>
> _***P78p Q12:*** It would also be good to add the missing notations to Fig. 2, e.g. the TPS parameter regressor S (line 152), M_theta and M_I (line 159), the warped features Y (line 161)._
>
> ***P78p A12:*** Thanks for the valuable suggestion, we agree with the reviewer that this modification will improve the diagram of Fig. 2. We will update the diagram with the missing notations.
>
> _***P78p Q13:*** What does the "TPS" in the green box in Fig. 2 stand for? Doesn't the TPS regressor S regress theta_tps directly?_
>
> ***P78p A13:*** No, the TPS in the green box is actually a fixed STN, which always samples $5\ times 5$ ResNet feature tensors centered on the coordinates of each keypoint. The only thing that changes is its affine matrix encoding the position of the keypoint. This sampled tensor is then forwarded to the TPS regressor network.
>
>
> _***P78p Q14:*** I don't understand the sentence in lines 186-188. Why is the regular (not the TPS-transformed) polar grid applied to the image (not just the patch)? What are the keypoint's attributes, where do they come from, have they been mentioned before? Why are they needed to get an identity transform? What is the difference between scale and size?_
>
> ***P78p A14:***  Thank you for the careful analysis and for pointing out that this part of the methodology is unclear. We agree that it is important to clarify this step of the proposed methodology as advised. Actually, the identity polar grid serves as the initialization to train the TPS model. The keypoint attributes comes from the keypoint detector. These attributes encode the keypoint location (its coordinates in the image), its canonical orientation (used to achieve image in-plane rotation invariance), and its size (an estimate of the support region around the keypoint that should used by the descriptor to describe it). Scale and size are actually the same parameter, we will rectify the mistake in the manuscript and include a more detailed description of the keypoint parameters.
> For example, let’s consider an input keypoint with size = 1.0, and coordinates (x = 100, y = 100) in the input image. The identity polar grid $\mathcal{G}_{I}$ in this case would be a polar grid with radius 0.5, centered at the coordinates (x = 100, y = 100). The TPS model requires this input grid of polar coordinates and the TPS parameters $\theta$ estimated by the regressor network to compute a transformed grid of coordinates $\mathcal{G}_\theta$. With this transformed grid, we can now sample a $32 \times 32$ image patch in the input image using bilinear interpolation, to account for both the keypoint parameters and the local deformations.
>
> _***P78p Q15:*** The sentence in lines 196-198 is a bit ambiguous and could be split into two shorter sentences, and the sentence in lines 198-199 could be pulled in front. "which" could refer to the loss or to HardNet. It is not obvious that HardNet comes from [22], only that the triplet loss is used from [22]._
>
> ***P78p A15:*** We agree with the reviewer and will split the sentence into two shorter ones. The term “which” refers to the loss. We will rephrase this part to improve its readability. Thank you for the careful reading of our paper.
>
> _***P78p Q16:*** A short explanation of what needed to change for adapting to R2D2 and ASLFeat (lines 282-283) in the appendix would be good for completeness._
>
> ***P78p A16:*** We agree with the reviewer, and we are adding this information in the supplementary material. Unfortunately, both ASLFeat and R2D2 did not allow the description for keypoints from different detectors (such as from SIFT detector). Therefore, to allow the comparison on closer common ground and conditions, we evaluated our descriptor using their detected keypoints (but this also required some adaptations, notably for the scale parameter). First, both ASLFeat and R2D2 do not provide keypoint orientation, which prevents using them as keypoints to most baselines. Fortunately, DEAL does not depend on orientation. ASLFeat also does not estimate or provides the detected keypoint’s scale. Therefore, we have empirically tested several scales to be used with our descriptor and chose the value of 2.5 for all keypoints (no keypoint orientation is required for DEAL). For R2D2 keypoints, we converted the keypoints’ scales to be SIFT-compatible by dividing them by 32.
>
> _***P78p Q17:*** Limitations are mentioned in one sentence (lines 371-372) but not discussed to give a good sense of them, and I am unable to locate the limitations that are supposed to be in the appendix. Qualitative results could be added._
>
> ***P78p A17:*** We agree with the reviewer that this is an important point. As discussed in the conclusions and for the applications shown in the supplementary material, the descriptor performance is compromised when dealing with surfaces containing sharp deformations (which also often results in occlusions) and poor keypoint detector repeatability. These conditions also hamper the competitors, as shown in Fig. 8 of the supplementary. Too strong deformations and low repeatability detectors can lead to poor description and correspondence of images. We highlight that we have briefly discussed the limitations of the description of poor textured surfaces in the tracking in lines 35-42 of the Supplem. Material. Two qualitative failure examples are also shown in Fig. 8, while matching images containing repetitive textured patterns and reflections. Following the reviewers’ suggestions, we are further discussing the limitations in the supplementary material.
>
> _***P78p Q18:*** Potential negative societal impact is not discussed. Better tracking of clothed people (which have non-rigid deformations) through surveillance might be one point._
>
> ***P78p A18:*** Thanks for this suggestion. We agree with the reviewer, and indeed the proposed descriptor could be used for recognizing and tracking humans from images, such as in surveillance contexts. Therefore, we are adding this information to the NeurIPS submission form checklist.

---

> > ### Comment · Reviewer_P78p · 2021-08-11
> > **Follow-up Questions**
> >
> > Thank you for taking the time to reply in detail. I have a few follow-up questions to the supplied answers.
> >
> > P78p Q11: The answer cannot be right because T_theta in line 159 is before Eq. 1 and does not have access to theta_tps, G_I or Eq. 2. I still do not understand the relationship between theta in line 159 (the STN section, incl. Equation 1), where it is used as T_theta(M_I), and theta in e.g. line 190 (the PTN section), where it is used as T_theta(G_I). If T_theta(G_I) in Fig. 2 does not use the theta from the STN section but instead uses theta_tps, then this should be changed in the figure. It is currently confusing since theta is used multiple times for different transformations. Or are all usages (lines 159, 190) intentionally the same transformation parameters theta? I suggest to use different letters for different transformations, I've spent a lot of time now trying to make sense of this and it will confuse any casual reader of the paper.
> >
> > P78p Q13: If the "TPS" in the green box in Fig. 2 does indeed what A13 states, then what is the first green box labeled "STN"? I am confused what the order of operations is.
> >
> > My understanding is this: (1) ResNet feature map X. (2) take the 5x5(x128) sub-tensor from X, centered on keypoint position (downscaled by factor 8). (3) resample that tensor with a non-rigidly deformed grid (not a straight grid) with the first STN with Eq. 1. (4) flatten. (5) feed into TPS regressor MLP S. (6) S outputs theta_tps. (7) theta_tps warps the identity polar grid into G_theta_tps with Eq. 2. (8) G_theta_tps is used to sample the image.
> >
> > If my understanding is correct, the top-right part of the figure needs some re-working: the right green box would be between steps (5) and (6), which doesn't make sense. If my understanding is incorrect, the method section needs clarifications because I've spent quite some time now trying to understand what's going on.
> >
> > Similarly for P78p Q6, I remain confused as to how step (5) can work well in principle if it only accesses deformed feature maps from step (3). But step (3) destroys some information because it's meant to normalize out some deformations, e.g.: let's say a leaf of a tree is centered at pixel (3,4) in an image A. Let's say the same leaf is recorded at a later time again with the same camera as image B and let's say it's now centered at pixel (3,6) because the wind is blowing stronger. It is conceivable that the output of step (3) is the exact same for both images because it is possible that the slight deformation of the leaf has been normalized out by the STN. Then the TPS regressor cannot distinguish between both cases and hence the same G_theta_tps will be used to sample the image A and to sample image B, even though the appearance of the leaf is different between both images. That seems like an issue to me.
> >
> > For P78p Q4, I also still do not follow. Why would skipping the first STN (step 3) lead to an identity polar grid? I could just take the 5x5x128 sub-tensor straight from X, flatten it, and give it to the TPS regressor S, i.e. simply skip step 3.
> >
> > I'd appreciate a linear list of simple steps as to what happens after what.

---

> > > ### Author Response · Authors · 2021-08-12
> > > **Reply: Follow-up Questions - Part I**
> > >
> > > _Thank you for taking the time to reply in detail. I have a few follow-up questions to the supplied answers._
> > >
> > > We sincerely appreciate the reviewer’s interest in our work. Furthermore, we are grateful for the reviewer bringing up this discussion to clarify what exactly happens to the transformations inside the network. There is indeed a confusing part in the diagram and text regarding the Spatial Transformers. We will revise it to allow the proper interpretation from readers and the re-phrasing of the paragraphs related to the Spatial Transformer.
> > >
> > > _***P78p Q11 (2):*** The answer cannot be right because T_theta in line 159 is before Eq. 1 and does not have access to theta_tps, G_I or Eq. 2. I still do not understand the relationship between theta in line 159 (the STN section, incl. Equation 1), where it is used as T_theta(M_I), and theta in e.g. line 190 (the PTN section), where it is used as T_theta(G_I). If T_theta(G_I) in Fig. 2 does not use the theta from the STN section but instead uses theta_tps, then this should be changed in the figure. It is currently confusing since theta is used multiple times for different transformations. Or are all usages (lines 159, 190) intentionally the same transformation parameters theta? I suggest to use different letters for different transformations, I've spent a lot of time now trying to make sense of this and it will confuse any casual reader of the paper._
> > >
> > > ***P78p A11 (2):*** Thanks to the reviewer for indicating a confusing description. In lines 156-162 we intend to introduce to the reader the Spatial Transformer Layer. Notice that the $\theta$ and $\mathbf{T_\theta}$ in line 158 defines a *generic* transformation (not a non-rigid one) that converts spatial coordinates of a reference 2D mesh grid into a target mesh grid, and the Sampler Layer (Equation 1) is used to sample those target coordinates in the input tensor for each channel. It is worth noting that the sampling is agnostic to the transformation used, since it only takes as input the transformed grid and the tensor to be sampled.
> > > The description becomes confusing since we introduced the concept of spatial transformers and described the use of the first STN in the same paragraph. We sincerely apologize for the confusing paragraph. To clarify to the reviewer, the parameter $\theta$ is used in the generic transformation $\mathbf{T_\theta}$ used to convert the spatial coordinates of an input grid $\mathcal{M_I}$ to transformed coordinates $\mathcal{M_\theta}$. On the other hand, the $\theta_{tps}$ are parameters computed by the regressor to be used in the TPS. We will re-work this paragraph, expliciting the $\theta_{tps}$ notation in Equation 2, fix the diagram of Figure 2, and separate the introduction of the generic Transformer Layer from the first STN used in our network. Specifically, we will make clear that the first STN is used to “differentiably crop” a (5,5,128) local tensor from ResNet features centered around each input keypoint spatial coordinates to convert it to the input features used for the TPS regressor network without deforming the original ResNet features.
> > > In order to make the diagram of Figure 2 clearer, we will: i) rename the block [PTN + TPS] to [PTN], since it is used to provide the identity polar grid $\mathcal{G_I}$ to the TPS warping function, and it only depends on the input keypoint parameters; ii) we will also explicitly indicate that the $\theta_{tps}$ comes directly from the regressor network by updating the diagram to show that $\mathbf{T_\theta}$ is referring to the transformation of the TPS (Equation 2).

---

> > > ### Author Response · Authors · 2021-08-12
> > > **Reply: Follow-up Questions - Part II**
> > >
> > > _***P78p Q13 (2):*** If the "TPS" in the green box in Fig. 2 does indeed what A13 states, then what is the first green box labeled "STN"? I am confused what the order of operations is.
> > > My understanding is this: (1) ResNet feature map X. (2) take the 5x5(x128) sub-tensor from X, centered on keypoint position (downscaled by factor 8). (3) resample that tensor with a non-rigidly deformed grid (not a straight grid) with the first STN with Eq. 1. (4) flatten. (5) feed into TPS regressor MLP S. (6) S outputs theta_tps. (7) theta_tps warps the identity polar grid into G_theta_tps with Eq. 2. (8) G_theta_tps is used to sample the image.
> > > If my understanding is correct, the top-right part of the figure needs some re-working: the right green box would be between steps (5) and (6), which doesn't make sense. If my understanding is incorrect, the method section needs clarifications because I've spent quite some time now trying to understand what's going on.
> > > Similarly for P78p Q6, I remain confused as to how step (5) can work well in principle if it only accesses deformed feature maps from step (3). But step (3) destroys some information because it's meant to normalize out some deformations, e.g.: let's say a leaf of a tree is centered at pixel (3,4) in an image A. Let's say the same leaf is recorded at a later time again with the same camera as image B and let's say it's now centered at pixel (3,6) because the wind is blowing stronger. It is conceivable that the output of step (3) is the exact same for both images because it is possible that the slight deformation of the leaf has been normalized out by the STN. Then the TPS regressor cannot distinguish between both cases and hence the same G_theta_tps will be used to sample the image A and to sample image B, even though the appearance of the leaf is different between both images. That seems like an issue to me_
> > >
> > > ***P78p A13 (2):*** Given the clarifications to the reviewer in the previous question (***P78p A11 (2)*** ), we provide a detailed linear list of steps from the input to the output of the entire model for an input image with a single keypoint for the sake of clarity.  Assume a tensor shape convention of (H=Height, W=Width, C=Channels). Let the input of the network be a (800,800,3) image with a single keypoint having parameters (x = 80, y = 80, size = 12.0) exactly at the deforming leaf on a tree the reviewer described.
> > > 1. The whole input image is forwarded to the ResNet-34 block, giving an output tensor $\mathbf{X}$ of shape (100, 100, 128).
> > > 2. The first STN takes the keypoint parameters (x=80, y=80) of the keypoint, and rescales those coordinates by 1/8, since these coordinates are in the original image and $\mathbf{X}$ is spatially downscaled by 1/8. Notice that the keypoint size is not considered in the first STN. The first STN samples a local patch of features of $\mathbf{X}$ with an affine matrix (the matrix encodes a translation of the keypoint position). The first STN then converts a $5\times 5$ identity mesh grid $\mathcal{M}_{I}$ (a mesh grid of $5\times5$ pixel coordinates centered at the origin) by translating it to become the transformed grid $\mathcal{M}_\theta$, a mesh grid centered at (10,10), i.e., at the position of the keypoint, by using the affine matrix. Finally, the Sampler Layer samples a local tensor (5,5,128) from $\mathbf{X}$ at the locations from $\mathcal{M}_\theta$. The resampling is performed for each keypoint; thus, each keypoint has a different local tensor if they lie in different spatial coordinates in the image. As we have only one keypoint in this example, there will be only one local tensor of shape (5,5,128). Please notice that in this step no deformation has been applied. We just resampled a local tensor from the ResNet-34 features centered at the position of the keypoint using bilinear interpolation. This local resampling is analogous to cropping a patch of size $5\times 5$ pixels in the original input image as done by classic hand-crafted local descriptors in Computer Vision such as SIFT or learned-based as TFeat and Log-Polar. However, in our network we use higher-level features coming from the ResNet feature maps, and differentiable sampling to perform the cropping task. Our experiments show evidence that instead of directly sampling the input image, the ResNet-34 is able to reason about higher-level contextual cues about the object, such as its shape, the scene illumination, and the relative configuration of the local texture of the object, to find the right local deformation parameters.
> > > 3. This is where the second STN begins. Both the Polar and TPS transformations are inside this STN block. The TPS regressor network MLP takes as input the flattened version of the “cropped” tensor from the previous layer ($5 \times 5 \times 128$) and generates as output $\theta_{tps}$, which is a parameter vector used by Equation 2.
> > > 4. The Polar Transformer (PTN) module generates an identity grid $\mathcal{G}_I$ using the keypoint attributes (x=80,y=80, size=12.0), *and only depends on the keypoint attributes*. The identity grid is created in the original image spatial coordinate space, therefore, for our single keypoint, the identity polar grid will be centered at coordinates (80,80) with radius of 12 pixels on the input image.
> > > 5. Using the TPS parameters $\theta_{tps}$ and $\mathcal{G_I}$ from the PTN, we apply Equation 2 to warp $\mathcal{G_I}$ into the deformed grid $\mathcal{G_\theta}$, which is then used by the Sampler Layer to sample pixels in the original image. The result is the rectified patch, which is expected to be invariant to local deformations. The invariance to deformations is enforced by the triplet loss in the training phase, since the network is fully differentiable and the NRW module learns meaningful deformations that increases the distinctiveness of the final descriptor.
> > > 6. At last, the rectified patch is forwarded to the HardNet that outputs a 128-D descriptor that will be robust to non-rigid deformations.
> > >
> > > In summary, if the network receives two different images with the single keypoint centered on the deforming tree leaf (as in the example given by the reviewer), the ResNet features will be different when the local shape, relative texture, and illumination change. As a result, for the same keypoint, $\theta_{tps}$ will be different for each input image. Therefore, the difference between the ResNet features extracted from the two input images will encode the relative local transformations that the TPS regressor needs, in order to sample the two patches in both defomed surfaces.
> > >
> > > _For ***P78p Q4***, I also still do not follow. Why would skipping the first STN (step 3) lead to an identity polar grid? I could just take the 5x5x128 sub-tensor straight from X, flatten it, and give it to the TPS regressor S, i.e. simply skip step 3._
> > >
> > > Thanks for the helpful observation. If we skip the first STN, it would be impossible to have the 5x5x128 sub-tensor. Considering the previous answer (***P78p A13 (2):***), if we do not have the first STN, there are two possible scenarios:
> > >
> > > 1. The first scenario is to feed the entire $\mathbf{X}$ tensor to the TPS regressor, but the number of parameters of the fully-connected layer would be very large; consequently, it would be extremely expensive for the considered input images. Moreover, if we do not use the first STN, the coordinates of the input keypoints would no longer be considered by the TPS regressor. Therefore, the NRW module will not be aware of the position of the keypoints, ultimately impairing the network to estimate a local deformation model for each keypoint.
> > > 2. The second scenario is what we tested in the ablation of the paper (Table 3 - PTN), where we no longer use both the ResNet-34 features and the regressor in the NRW-module. This scenario results in using the identity polar grid to re-sample the pixels from the images. This is akin to the method Log-Polar.

---

> > > > ### Comment · Reviewer_P78p · 2021-08-22
> > > > **Thank you**
> > > >
> > > > Thank you for the thorough explanation and for taking the time for it. I now have a good understanding of the method.

---

### Official Review · Reviewer_1fX6 · 2021-07-16

**Rating:** 7
**Confidence:** 3

**Summary:**

The paper introduces a new deep learning pipeline for the computation of local image features that are robust to non-rigid deformations. The proposed method is end-to-end, and works by simulating non-rigid transformations on a synthetic dataset to jointly learn how to rectify the deformed patches, while extracting discriminative and invariant features at the same time. Comparisons with several state of the art pipelines on multiple datasets confirm the effectiveness of the proposed approach.

**Limitations And Societal Impact:**

The limitations are briefly discussed in the conclusions, and failure cases are included in the supplementary material.

**Main Review:**

I enjoyed reading the paper, which is well written, fluent, and easily accessible for someone not fully familiar with this family of methods (such as this reviewer). The method is described well and the computational modeling is sound. The underlying idea of explicitly parametrizing deformations via a learnable TPS model is straightforward and well executed, and I believe there is value in this simplicity, given the good results it enables in several practical cases.

Worth mentioning is the extensive experimental evaluation, which includes several recent datasets comprised of real-world deformable objects (despite the training being based on synthetic examples) undergoing diverse picture-space transformations (e.g. illumination, viewpoint, object deformation), as well as comparisons with several baselines and recent learning-based methods. Hyperparameter sensitivity, ablation, and runtime studies are also included. Two real-world applications are analyzed, demonstrating competitive results if compared to other recent methods.

As a minor comment, I found the results reported in Table 3 (Ablation) a bit difficult to interpret: if the NRW component is key to learning the unwrap, why is the improvement so mild over vanilla PTN (+0.04 mean matching accuracy)? The paper refers to this as significant improvement, but more comments on this being a true gain would be useful.

Overall, I believe this is a solid paper that deserves to be communicated. At the same time, I have some reservations on this being a computer vision paper first and foremost, with machine learning just being a convenient ingredient to compose the full pipeline. In other words, I think the key contribution is in the vision area, while it might be not very significant for ML. Therefore I am not 100% confident NeurIPS is the right venue for it, but I won't oppose acceptance based solely on this doubt.

**Time Spent Reviewing:**

4

---

> ### Author Response · Authors · 2021-08-10
> **Point-by-point reply to Reviewer 1fX6**
>
> _I enjoyed reading the paper, which is well written, fluent, and easily accessible for someone not fully familiar with this family of methods (such as this reviewer). The method is described well and the computational modeling is sound. The underlying idea of explicitly parametrizing deformations via a learnable TPS model is straightforward and well executed, and I believe there is value in this simplicity, given the good results it enables in several practical cases._
>
> We sincerely appreciate all the valuable feedback.
>
> _Worth mentioning is the extensive experimental evaluation, which includes several recent datasets comprised of real-world deformable objects (despite the training being based on synthetic examples) undergoing diverse picture-space transformations (e.g. illumination, viewpoint, object deformation), as well as comparisons with several baselines and recent learning-based methods. Hyperparameter sensitivity, ablation, and runtime studies are also included. Two real-world applications are analyzed, demonstrating competitive results if compared to other recent methods._
>
> We are grateful to the reviewer’s positive comments and effort dedicated to the reading of our paper.
>
> _***1fX6 Q1:*** As a minor comment, I found the results reported in Table 3 (Ablation) a bit difficult to interpret: if the NRW component is key to learning the unwrap, why is the improvement so mild over vanilla PTN (+0.04 mean matching accuracy)? The paper refers to this as significant improvement, but more comments on this being a true gain would be useful._
>
> ***1fX6 A1:*** We appreciate the reviewer's comments and the thoughtful suggestion. We will improve the discussion of this experiment in the paper to better reflect the true gain in matching scores gains achieved by our proposed NRW component. The reported value on Table 3 actually means that the NRW component alone was able to improve the matching accuracy by 4% in comparison to using the vanilla PTN on a challenging deforming sequence.
> It may not seem a large improvement at first glance; however, the inlier ratio can greatly affect the performance of Random Sample Consensus (RANSAC)-based robust registration methods, increasing the number of iterations required to achieve a certain probability that a set of inlier points will be sampled from the correspondences to estimate a model. For instance, if one needs a minimal sample of $n$ inlier points to estimate a model with probability $p$, one can estimate the number of iterations $k$ using the following equation:
>
> $k$ = $log(1 - p ) / log(1 - r^n)$,
>
> where $r$ is the inlier ratio, i.e., the ratio of the correct points relative to the total number of points. For example, if one requires a minimal sample of $n = 30$, setting the probability of success $p=0.99$ that the model will be correct, and considering inlier ratios $r = 0.70$ and $0.74$, substituting in the above equation, we have that $k = 204,315$ for $r = 0.70$ and $k = 38,572$ for $r = 0.74$, meaning that, by using our NRW component, one requires only about 18% of of the number of iterations required (i.e. more than 5x performance gains) to robustly fit a model to those points when compared to not using the NRW.

---

### Official Review · Reviewer_fuQg · 2021-07-16

**Rating:** 7
**Confidence:** 4

**Summary:**

This paper proposes to learn deformation-aware local feature descriptors by rectifying the local region with carefully designed spatial transformer networks (STNs). Instead of using 3D training data, the paper proposes to learn from deformed 2D samples. The feature extraction network utilizes two STNs to rectify the local region and HardNet to extract the final feature. Among the two STNs, the second STN is specially designed with the thin-plate-spline transform and a polar grid so that the resulting region flexibly deform from a circular shape. The experimental results show that better or competitive results with much less processing time.

**Limitations And Societal Impact:**

Limitations are described in the final section.

**Main Review:**

+- Not highly novel, but practically beneficial: Using spatial transformers and deep learning to learn robust features is not exactly new. I think that the main contributions are in designing it for general deformation, and also in the detailed design (a simple but specially designed deformation module with two STNs) of the proposed method. On the other hand, the method has many practical benefits: no requirement for 3D training data, fast processing time, and good performance.

-- Missing reference: There is a highly related work that uses STN for learning local features [Ono et al., NeurIPS 2018], which is missing in the current paper. This has to be mentioned and compared with the proposed method.

-- Why do we need two STNs?: I said that one of the main contributions is the module with two STNs. But why do we need this? Can't we just use the second STN only? What is the exact role of the first (plain) STN? The reason for such a design is not well explained in the paper. I strongly suggest providing (or analyzing) this reason, which can be crucial in lifting the value of the paper.

+- Using only 2D data: Although I mentioned that requiring only 2D data is a benefit, there is an issue to discuss here. I see that the proposed network can also be trained with RGB-D data with no problem. (In this case, moreover, we might also add some more supervisions, say, something like deformation supervision.) What happens if we do that? For example, the proposed method and GeoBit are competitive in performance in the experiments. Do things get any different with RGB-D data?

[After rebuttal]
Most of my concern has been addressed in the authors' response. I believe that this is a sound contribution in the field. I raise my score to 7.

**Time Spent Reviewing:**

3

---

> ### Author Response · Authors · 2021-08-10
> **Point-by-point reply to Reviewer fuQg**
>
> ***fuQg Q1:*** +- _Not highly novel, but practically beneficial: Using spatial transformers and deep learning to learn robust features is not exactly new. I think that the main contributions are in designing it for general deformation, and also in the detailed design (a simple but specially designed deformation module with two STNs) of the proposed method. On the other hand, the method has many practical benefits: no requirement for 3D training data, fast processing time, and good performance._
>
> ***fuQg A1:*** Thanks for the encouraging words and the supportive comments about our work.
>
> _***fuQg Q2:*** -- Missing reference: There is a highly related work that uses STN for learning local features [Ono et al., NeurIPS 2018], which is missing in the current paper. This has to be mentioned and compared with the proposed method._
>
> ***fuQg A2:*** We thank the reviewer for the suggestion. Following the reviewer's suggestion, we have updated the related work and we  included LFNet [Ono et al., NeurIPS 2018] in the experimental section (see Table: Comparison with state-of-the-art descriptors.). Please notice that we have tried to accommodate your suggestion and those recommendations by other reviewers by adding other descriptors in our comparison (SOSNet, R2D2, SIFT, ASLFeat, D2-Net, LIFT) to the Table: Comparison with state-of-the-art descriptors.
>
> ***Table: Comparison with state-of-the-art descriptors.*** The ones marked with * are computed on RGB-D images, and methods with ** were re-trained using our proposed non-rigid dataset. Results indicate that our proposed descriptor achieves state-of-the-art performance for matching deformable surfaces. In addition, our proposed dataset can also improve the robustness of existing methods to non-rigid deformations. Results are sorted by increasing MMA.
>
> |Method|Kinect1|Kinect2Sampled|DeSurTSampled|SimulationICCV| Avg. MS / MMA ↑|
> |:-----|-----:|-----:|-----:|-----:|-----:|
> |BRAND*|0.17 / 0.34|0.22 / 0.49|0.14 / 0.33|0.04 / 0.09|0.16 / 0.34|
> |R2D2|0.17 / 0.36|0.25 / 0.59|0.14 / 0.32|0.06 / 0.16|0.17 / 0.39|
> |ORB|0.19 / 0.39|0.25 / 0.55|0.18 / 0.40|0.14 / 0.30|0.20 / 0.43|
> |SOSNet|0.17 / 0.34|0.25 / 0.55|0.17 / 0.38|0.26 / 0.57|0.22 / 0.47|
> |DAISY|0.23 / 0.47|0.29 / 0.62|0.16 / 0.37|0.19 / 0.39|0.22 / 0.48|
> |FREAK|0.24 / 0.49|0.33 / 0.72|0.16 / 0.38|0.15 / 0.31|0.23 / 0.51|
> |SIFT|0.20 / 0.42|0.26 / 0.57|0.19 / 0.43|0.27 / 0.59|0.23 / 0.51|
> |DaLI|0.25 / 0.51|0.35 / 0.76|0.21 / 0.48|0.10 / 0.22|0.25 / 0.54|
> |TFeat**|0.23 / 0.48|0.28 / 0.62|0.20 / 0.46|0.28 / 0.61|0.25 / 0.55|
> |TFeat|0.25 / 0.50|0.28 / 0.61|0.21 / 0.48|0.29 / 0.63|0.26 / 0.56|
> |ASLFeat|0.31 / 0.58|0.39 / 0.69|0.28 / 0.53|0.19 / 0.35|0.31 / 0.56|
> |Log-Polar|0.28 / 0.58|0.30 / 0.65|0.23 / 0.54|0.22 / 0.49|0.26 / 0.57|
> |D2-Net|0.20 /0.50 |0.23 / 0.82| 0.14 / 0.47|0.11 / 0.30|0.17 / 0.57|
> |LFNet|0.44 / 0.40|0.51 / 0.43|0.28 / 0.77|0.21 / 0.74|0.36 / 0.59|
> |LIFT|0.09 / 0.57|0.16 / 0.65|0.08 / 0.52|0.13 / 0.73|0.12 / 0.62|
> |LogPolar**|0.29 / 0.60|0.31 / 0.69|0.24 / 0.56|0.33 / 0.72|0.29 / 0.65|
> |GeoBit*|0.31 / 0.65|0.35 / 0.77|0.20 / 0.47|0.32 / 0.71|0.30 / 0.66|
> |Ours|0.33 / 0.68|0.38 / 0.85|0.27 / 0.63|0.36 / 0.80|0.34 / 0.75|
>
> _***fuQg Q3:*** -- Why do we need two STNs?: I said that one of the main contributions is the module with two STNs. But why do we need this? Can't we just use the second STN only? What is the exact role of the first (plain) STN? The reason for such a design is not well explained in the paper. I strongly suggest providing (or analyzing) this reason, which can be crucial in lifting the value of the paper._
>
> ***fuQg A3:***  Thanks to the reviewer for indicating this unclear description. We agree that it is important to clarify the role of the first STN in our network. The first STN is used to interpolate a $5 \times 5$ tensor coming from the ResNet-34 feature maps centered at the coordinates of each input keypoint in the image (downscaled to ⅛). This can be seen on the diagram of Fig. 2 in the paper (small red dots in the full tensor $\mathbf{X}$. For example, if there exists a keypoint at position (x = 80, y = 80) in the input image, the first STN would re-sample a $5 \times 5$ local tensor of shape (5,5,128) at position (x = 10, y = 10) of the feature map coming from ResNet. This is done for each input keypoint. The local feature map encodes mid-level image features and is fed to the localization network (a fully-connected MLP) to estimate the TPS warping parameters. In other words, the first STN is used to sample the ResNet-34 mid-level feature maps that are used by the deformation module. The TPS regression estimates the transformation to be applied, and the sampling strategy would be equivalent to a fixed polar sampling of the patch centered at the keypoint coordinates with radius proportional to its size if the TPS is not applied. The ablation study of using or not this first STN + TPS  can be seen in Table 3 of the paper (PTN vs NRW). This strategy plays an important role in capturing the non-rigid deformations as shown in the qualitative examples and in the new results of Table: Comparison with state-of-the-art descriptors, where we re-trained Log-Polar descriptor using our non-rigid dataset. We will improve the discussion about the importance of the first STN in the paper.
>
> ***fuQg Q4:*** +- _Using only 2D data: Although I mentioned that requiring only 2D data is a benefit, there is an issue to discuss here. I see that the proposed network can also be trained with RGB-D data with no problem. (In this case, moreover, we might also add some more supervisions, say, something like deformation supervision.) What happens if we do that? For example, the proposed method and GeoBit are competitive in performance in the experiments. Do things get any different with RGB-D data?_
>
> ***fuQg A4:*** We would like to thank the reviewer for bringing this matter up. We do agree that methods such as GeoBit and BRAND that are able to process RGB-D data can play a central role in data acquired using RGB-D devices such as Microsoft Kinect. However, working on RGB images benefits a wider range of applications depending on legacy visual data, e.g., monocular videos and RGB images on the web. Therefore, we established that only visual data will be used in the inference step in our designing principles.  We agree with the reviewer that feeding our network with more information in training could be beneficial. We did several tests using depth data to improve the results or to accelerate the convergence of the optimization. Still, we have noted that adding depth in our architecture as the way it was conceived is not straightforward and can make the optimization unstable. Specifically, since the sampling is performed in the image domain and RGB-D data are noisy and have missing in many regions (i.e., holes in the depth map), the non-rigid warper component becomes unable to find the transformation that can simultaneously provide a good image rectification and be consistent with depth data. It is worth mentioning that the warping transform should provide a good image rectification to enable the feature extractor network and not necessarily to concur with the depth data.

---

### Official Review · Reviewer_dECi · 2021-07-19

**Rating:** 5
**Confidence:** 3

**Summary:**

This paper presents a learnt local keypoint descriptor for deformable objects. It is trained in a supervised fashion using the HardNet network and loss on the SIFT keypoints of a synthetic dataset (succinctly presented in the paper, based on existing SfM datasets). The novelty is the addition of a spatial transformer network (STN) to normalize the patches, inline with recent local descriptor learning approaches. As far as I understand the main conceptual difference with existing works is that the (STN) is not only affine but uses a TPS on a log-polar grid. My understanding of the paper is that it argues this part is crucial for matching deformable objects.

**Limitations And Societal Impact:**

.discussion of limitations and failure cases missing

**Main Review:**

Novelty/contributions/ablation:
While the proposed approach makes sense, I do not really see any novelty/important difference compared to standard learned keypoint descriptors except (i) that the deformation learned by the transformer network includes a TPS deformation on a log-polar grid (and not only an affine one) (ii) the learning dataset seem to be an "in house" dataset (on which I would appreciate much more details and which I would like the authors to confirm they will release with their code) with a HardNet approach. It is unclear to me what is the influence of each on the results. There is only an embryo of ablation study (Table 3, two numbers on an unspecified handpicked small subset of the data) where the full transformer network is ablated. I would like to also see **an ablation of only the TPS part, and a TPS on a standard regular grid**, to also have access to the corresponding results on the full dataset, and to have some analysis of the effect of the training dataset (e.g. same data generation process but without the non-rigid deformations, size). More generally, I would also like to see ablation that replace different parts of the proposed pipeline (loss, network archi, training data) with those of a recent learned feature descriptor.

Comparisons to other approaches:
My main concern is that the results are reported on two recent (2019) datasets that have been used by very few other works (5 and 2 citations), one thus has to rely on the paper itself to run relevant baselines. While there is a clear effort in this direction in Table 2 and 4, there are some baselines that seem important to me and would be worth adding a few: SIFT, LIFT, superpoint, D2-Net, R2D2 and maybe other recent learnt descriptors. Note that a single number on R2D2 is reported in 4.2, but I think this is not really sufficient and it should be compared with other methods (even if it makes the table more complicated because they can be used either with SIFT keypoints or with their own keypoints) Conversely, I would like to see the performance of the proposed descriptor on standard keypoint matching benchmarks (potentially for downstream tasks such as calibration). While I understand the advantage of the proposed keypoints is to handle deformable objects, they should also be able to handle rigid object, and showing they give competitive results on standard benchmark would reassure me that the results reported in the paper are indeed impressive and not only due to comparison bias.

In conclusion, there are too many things in this paper I am not confident about and that I would like to see added to recommend acceptance. I see little demonstrated important technical and conceptual novelty and I don't think improving SoA on seldom used datasets is itself sufficient for a publication.

Minor: some writing issue, e.g. last sentence of the first paragraph "After all, we live in a non-rigid world, where beyond rigid surfaces, it is also populated ...'


**Time Spent Reviewing:**

2

---

> ### Author Response · Authors · 2021-08-10
> **Point-by-point reply to Reviewer dECi - Part I**
>
> _***dECi Q1:*** Novelty/contributions/ablation: While the proposed approach makes sense, I do not really see any novelty/important difference compared to standard learned keypoint descriptors except (i) that the deformation learned by the transformer network includes a TPS deformation on a log-polar grid (and not only an affine one) (ii) the learning dataset seem to be an "in house" dataset (on which I would appreciate much more details and which I would like the authors to confirm they will release with their code) with a HardNet approach._
>
> ***dECi A1:*** We thank the reviewer for the careful analysis of our paper. The main contributions of our work reach beyond the use of a TPS model.  In fact, our model not only replaces the affine transformation with a TPS deformation model, but the proposed architecture allows the network to reason about image deformations, taking into account both low-level local features and the global context of the deformation affecting the full image. This is a major advance, and it provides awareness of the transformations affecting the patch in several ways, notably when dealing with keypoints in poorly textured regions or with ambiguous texture patterns. For recalling the components in our architecture allowing this awareness are:
>
> (i) A ResNet-34 branch that is aware of the entire image, allowing higher-level contextual cues to be used by the localization network;
> (i) TPS warper module, which for the best of our knowledge, is the first learned local descriptor method to leverage the representativeness of thin-plate-splines to model local deformations;
> (ii) An enhanced Polar Transformer, enabling the network to be rotation equivariant. It is important to notice that we extend the idea of Log-Polar’s original work, explicitly leveraging the rotation equivariance property to achieve full rotation invariance on descriptor space.
>
> As shown in the provided experiments, the approach improved the description and correspondence capabilities over different datasets. We argue the combination of the components in our architecture working together for extracting local image descriptors with global transformation awareness, in addition to our proposed deformation dataset, all of which we understand to be of relevance and contribution to be archived in that it brings relevant insights in the study of matching deformation objects for the community.
>
> We confirm that we will release the architecture and training code, including the dataset used for training. We also agree with the reviewer that the training dataset implementation details are missing, and we will update the paper and the supplementary material with the following paragraph, complementing the existing training dataset paragraph:
>
> Our physics simulation framework is implemented in OpenGL, achieving the necessary efficiency and flexibility to allow low-cost generation of thousands of images of realistic deforming objects suitable for training Deep Neural Networks, permitting efficient low-level access to the simulation data, i.e.,  Z-buffer, camera parameters, and perfect correspondence in the sequences. The objects are represented by a grid of particles having mass in 3D space. Deformations are induced by the forces applied onto them, implemented as wind and gravity.  A constraint satisfaction optimization step is performed over all particles to enforce a constant distance of neighboring particles, thus keeping the deformation isometric. For each simulation round, we generate (i) random wind forces in all directions to generate diverse object deformations in a chaotic fashion, (ii) illumination variation such as intensity, global position, number of light sources, directional lighting, and color changes to enforce realistic non-linear illumination diversity, and (iii) Gaussian noise in image pixels to simulate real camera sensors.
>
>
> _***dECi Q2:*** It is unclear to me what is the influence of each on the results. There is only an embryo of ablation study (Table 3, two numbers on an unspecified handpicked small subset of the data) where the full transformer network is ablated. I would like to also see an ablation of only the TPS part, and a TPS on a standard regular grid, to also have access to the corresponding results on the full dataset, and to have some analysis of the effect of the training dataset (e.g. same data generation process but without the non-rigid deformations, size). More generally, I would also like to see ablation that replace different parts of the proposed pipeline (loss, network archi, training data) with those of a recent learned feature descriptor_
>
>
> ***Table: Ablation.***
>
> | Method | MMA ↑|
> |:---------|-------:|
> |Cartesian NRW| 0.68|
> |Planar Data NRW| 0.75|
> |PTN-only| 0.75
> |Polar NRW| 0.79 |
>
>
> ***dECi A2:*** We are thankful for the valuable suggestions, which will surely increase the quality of our paper. As kindly suggested by the reviewer, we have performed two ablations, namely, the use of the TPS on a standard regular grid patch and training the proposed architecture on a new dataset containing only rigid planar objects shown in Table: Ablation. The new ablation tests show that (i) the PTN sampler is able to improve matching accuracy, (ii) the non-rigid dataset also helps to improve accuracy, and finally (iii) the NRW component also boosts performance. We will update Table 3 of the paper with those new ablation tests. We also analyzed the effect of training the learned descriptors Log-Polar (which is akin to our method without the TPS module) and TFeat on the same training data we used to train our architecture. The state-of-the-art comparison table in the paper was updated with other recent baselines in addition to the re-trained methods and can be checked in Table: Comparison with state-of-the-art descriptors, which contains the results in the entire database, as requested by the reviewer. Regarding the subset used in the ablation studies specifically for the non-rigid warper (NRW) module, it is worth mentioning that we selected the sequence containing the most challenging deformations in order to better assess the NRW influence in matching accuracy on an edge case, taking into consideration that several sequences have weak deformation overall, particularly on the Kinect1 dataset.
>
> ***Table: Comparison with state-of-the-art descriptors.*** The methods marked with * are computed on RGB-D images, and methods with ** were re-trained using our proposed non-rigid dataset. Results indicate that our proposed descriptor achieves state-of-the-art performance for matching deformable surfaces overall. In addition, our proposed dataset can also improve the robustness of existing methods to non-rigid deformations. Results are sorted by increasing MMA scores.
>
> |Method|Kinect1|Kinect2Sampled|DeSurTSampled|SimulationICCV| Avg. MS / MMA ↑|
> |:-----|-----:|-----:|-----:|-----:|-----:|
> |BRAND*|0.17 / 0.34|0.22 / 0.49|0.14 / 0.33|0.04 / 0.09|0.16 / 0.34|
> |R2D2|0.17 / 0.36|0.25 / 0.59|0.14 / 0.32|0.06 / 0.16|0.17 / 0.39|
> |ORB|0.19 / 0.39|0.25 / 0.55|0.18 / 0.40|0.14 / 0.30|0.20 / 0.43|
> |SOSNet|0.17 / 0.34|0.25 / 0.55|0.17 / 0.38|0.26 / 0.57|0.22 / 0.47|
> |DAISY|0.23 / 0.47|0.29 / 0.62|0.16 / 0.37|0.19 / 0.39|0.22 / 0.48|
> |FREAK|0.24 / 0.49|0.33 / 0.72|0.16 / 0.38|0.15 / 0.31|0.23 / 0.51|
> |SIFT|0.20 / 0.42|0.26 / 0.57|0.19 / 0.43|0.27 / 0.59|0.23 / 0.51|
> |DaLI|0.25 / 0.51|0.35 / 0.76|0.21 / 0.48|0.10 / 0.22|0.25 / 0.54|
> |TFeat**|0.23 / 0.48|0.28 / 0.62|0.20 / 0.46|0.28 / 0.61|0.25 / 0.55|
> |TFeat|0.25 / 0.50|0.28 / 0.61|0.21 / 0.48|0.29 / 0.63|0.26 / 0.56|
> |ASLFeat|0.31 / 0.58|0.39 / 0.69|0.28 / 0.53|0.19 / 0.35|0.31 / 0.56|
> |Log-Polar|0.28 / 0.58|0.30 / 0.65|0.23 / 0.54|0.22 / 0.49|0.26 / 0.57|
> |D2-Net|0.20 /0.50 |0.23 / 0.82| 0.14 / 0.47|0.11 / 0.30|0.17 / 0.57|
> |LFNet|0.44 / 0.40|0.51 / 0.43|0.28 / 0.77|0.21 / 0.74|0.36 / 0.59|
> |LIFT|0.09 / 0.57|0.16 / 0.65|0.08 / 0.52|0.13 / 0.73|0.12 / 0.62|
> |LogPolar**|0.29 / 0.60|0.31 / 0.69|0.24 / 0.56|0.33 / 0.72|0.29 / 0.65|
> |GeoBit*|0.31 / 0.65|0.35 / 0.77|0.20 / 0.47|0.32 / 0.71|0.30 / 0.66|
> |Ours|0.33 / 0.68|0.38 / 0.85|0.27 / 0.63|0.36 / 0.80|0.34 / 0.75|
>
> _***dECi Q3:*** Comparisons to other approaches: My main concern is that the results are reported on two recent (2019) datasets that have been used by very few other works (5 and 2 citations), one thus has to rely on the paper itself to run relevant baselines. While there is a clear effort in this direction in Table 2 and 4, there are some baselines that seem important to me and would be worth adding a few: SIFT, LIFT, superpoint, D2-Net, R2D2 and maybe other recent learnt descriptors. Note that a single number on R2D2 is reported in 4.2, but I think this is not really sufficient and it should be compared with other methods (even if it makes the table more complicated because they can be used either with SIFT keypoints or with their own keypoints)_
>
> ***dECi A3:*** Thanks for the valuable suggestions. We certainly agree with the reviewer about the benefits of having more datasets to evaluate the descriptor's performance. However, unfortunately, providing ground truth matches with suitable accuracy for non-rigid deformations is a hard task. Unlike rigid transforms, where we can easily compute a homography to map pixels from one image to another, a non-rigid transform must be computed for each kind of deformation and change in the camera's viewpoint. Nevertheless, we highlight that both datasets provide challenging samples with strong deformation, illumination changes, and point-of-view and sub-pixel match accuracy. As kindly suggested, we included in our experiments suggested baselines (SIFT, R2D2, LFNet, SOSNet, ASLFeat, D2-Net, LIFT) using SIFT keypoints when possible, or their own keypoints otherwise, as shown in Table: Comparison with state-of-the-art descriptors.

---

> ### Author Response · Authors · 2021-08-10
> **Point-by-point reply to Reviewer dECi - Part II**
>
> _***dECi Q4:*** Conversely, I would like to see the performance of the proposed descriptor on standard keypoint matching benchmarks (potentially for downstream tasks such as calibration). While I understand the advantage of the proposed keypoints is to handle deformable objects, they should also be able to handle rigid object, and showing they give competitive results on standard benchmark would reassure me that the results reported in the paper are indeed impressive and not only due to comparison bias._
>
> ***dECi A4:*** We thank the reviewer for suggesting the performance evaluation of our descriptor for also matching images of rigid objects. We selected the widely adopted HPatches dataset containing predominantly rigid objects but under severe viewpoint and illumination changes. The dataset is composed of images from 116 planar scenes with 5 pairs of images each (we selected the "Full image sequences"). The dataset is split between pairs affected either by strong viewpoint or illumination changes, and the results are presented independently for each split (following the protocol presented in LogPolar, we considered 51 scenes in the illumination split or 50 in the viewpoint split). We detected 1k SIFT keypoints independently for each image, which are used for all descriptors except for ASLFeat, LFNet (suggested by reviewer fuQg) and R2D2, which detect their own 1k keypoints.  For the evaluation, we adopted the matching score (MS) metric. The illumination and viewpoint splits have 255 and 250 pairs of images, respectively.
>
> ***Table: HPatches benchmark.*** Matching Scores on illumination and view split of the HPatches dataset. Our method achieves competitive performance among the methods using SIFT keypoints, and it has the second best results using ASLFeat keypoints.
>
> |Rank|Desc | MS view. | MS illu. | MS avg. |
> |:----|:-------|---------:|-----------:|----------:|
> |1|ASL | 0.34 | 0.44 | 0.39 |
> |2|Ours (ASL-kpts) | 0.33 | 0.35 | 0.34 |
> |3|R2D2 | 0.26 | 0.38| 0.32 |
> |4|Ours | 0.26 | 0.23 | 0.25 |
> |5|TFeat | 0.26 | 0.24 | 0.25 |
> |6|SIFT | 0.24 | 0.23 | 0.23 |
> |7|LFNet | 0.20 | 0.23 | 0.22 |
> |8|Log-Polar| 0.09 | 0.29 | 0.19 |
>
> As recommended, these experiments on HPatches demonstrate that the proposed descriptor capabilities also display competitive results to images without deformed objects using the keypoints detected by SIFT and the best performing method (ASLFeat). Table: HPatches benchmark  presents the matching scores in the illumination and viewpoint splits, as well as the average; our method is close to the recent state-of-the-art descriptors, specifically designed and trained for matching rigid objects (images affected by homography). Those methods were also trained with real images. Worth mentioning is the fact that R2D2 and ASLFeat are not invariant to image in-plane rotations; thus, in this dataset, our descriptor is doubly penalized by its invariance to transformations that do not exist in HPatches. Nevertheless, as demonstrated, our descriptor achieves competitive performance among the descriptors using the same set of keypoints (TFeat, Log-Polar, SIFT), which are specifically designed to work with planar scenes and were trained on real-world images.
>
> ***dECi Q5:*** _Minor: some writing issue, e.g. last sentence of the first paragraph "After all, we live in a non-rigid world, where beyond rigid surfaces, it is also populated ..._
>
> ***dECi A5:*** Thanks to the reviewer for indicating this poorly written sentence. We have corrected and completed the phrase in the revised manuscript. We also performed careful text proofreading.
>
> _***dECi Q6:*** Limitations And Societal Impact: discussion of limitations and failure cases missing_
>
> ***dECi A6:*** Thanks for pointing this out. A limiting factor is the presence of extreme deformation on a surface. In this case, even if our networks could correctly find the warping transform and correctly rectify the input image, there would be a loss of data due to the interpolation of sub-sampled pixels from the image. We have added in the supplementary material challenging sequences with objects producing failure cases. We highlighted and added the discussion in a more explicit form in the paper to clarify this point.

---

### Author Response · Authors · 2021-08-10
**Reply to all reviewers**

We deeply thank all the reviewers for their insightful comments and careful reviewing of our paper. We really appreciate the efforts and time spent to elaborate and write detailed suggestions that will significantly improve the quality of our work. Following the advice of the reviewers, if accepted, in the final version of the paper, we will highlight the following novel aspects:
* We have re-trained Log-Polar and TFeat descriptors using our non-rigid dataset, to clarify the raised concerns about the proposed non-rigid warper (NRW) module and its impact on the matching performance. Please check _Table: Comparison with state-of-the-art descriptors_ in the replies .
* A new experiment on the widely adopted HPatches [1] dataset for matching planar scenes is also provided (See _Table: HPatches benchmark_ in the following comments). In these experiments, we compare our method to the state-of-the-art local feature descriptors ASLFeat, R2D2, Log-Polar, TFeat, and SIFT. We plan to include these results in the supplementary material.
* We perform additional ablation tests ( _Table: Ablation_ in the replies), varying the training data and additional NRW parameters, to better evaluate the contribution of the key components of our proposed method, designed to solve the task of description of images with deformable objects.
* Several additional competitors are included following the reviewers’ suggestions. An improved version of Table 2 of the paper, containing the suggested additional baselines (SIFT, R2D2, LFNet, SOSNet, ASLFeat, D2-Net, and LIFT) using SIFT keypoints when possible, or their own keypoints otherwise. The results are shown in the following comments in _Table: Comparison with state-of-the-art descriptors._

[1] Balntas, Vassileios, et al."HPatches: A benchmark and evaluation of handcrafted and learned local descriptors." Proceedings of the IEEE conference on computer vision and pattern recognition. 2017.

---

### Decision · Program_Chairs · 2021-09-27

**Decision:**

Accept (Poster)

**Comment:**

This paper proposes an approach to learn deformation-aware local feature descriptors using spatial transformer networks (STNs).  The reviewers found the paper clear, the method practically useful and the experimental comparison to be comprehensive with the results quite strong.    I agree with this assessment.  I also appreciate the robust and healthy discussion between the reviewers and the authors that I think helped point in directions that make the paper better.